# Using artificial neural networks to predict riming from Doppler cloud radar observations

Teresa Vogl[1], Maximilian Maahn[1], Stefan Kneifel[2], Willi Schimmel[1], Dmitri Moisseev[3,4], and Heike Kalesse-Los[1]

[1]Institute for Meteorology, University of Leipzig, Leipzig, Germany
[2]Institute for Geophysics and Meteorology, University of Cologne, Cologne, Germany
[3]Institute for Atmospheric and Earth System Research/ Physics, Faculty of Science, University of Helsinki, Finland
[4]Finnish Meteorological Institute, Helsinki, Finland

**Correspondence:** Teresa Vogl (teresa.vogl@uni-leipzig.de)

**Abstract.** Riming, i.e. the accretion and freezing of supercooled liquid water (SLW) on ice particles in mixed-phase clouds, is an important pathway for precipitation formation. Detecting and quantifying riming using ground-based cloud radar observations is of great interest, however, approaches based on measurements of the mean Doppler velocity (MDV) are unfeasible in convective and orographically influenced cloud systems. Here, we show how artificial neural networks (ANNs) can be used to predict riming using ground-based zenith-pointing cloud radar variables as input features. ANNs are a versatile means to extract relations from labeled data sets, which contain input features along with the expected target values. Training data are extracted from a data set acquired during winter 2014 in Finland, containing both Ka- and W-band cloud radar and in-situ observations of snowfall by a Precipitation Imaging Package, from which the rime mass fraction ($\text{FR}_{PIP}$) is retrieved. ANNs are trained separately either on the Ka-band radar or the W-band radar data set to predict the rime fraction $\text{FR}_{ANN}$. We focus on two configurations of input variables: ANN #1 uses the equivalent radar reflectivity factor (Ze), MDV, the width from left to right edge of the spectrum above the noise floor (spectrum edge width; SEW), and the skewness as input features. ANN #2 only uses Ze, SEW and skewness. The application of these two ANN configurations to case studies from different data sets demonstrates that both are able to predict strong riming ($\text{FR}_{ANN} > 0.7$) and yield low values ($\text{FR}_{ANN} \leq 0.4$) for unrimed snow. In general, the predictions of ANN #1 and ANN #2 are very similar, advocating the capability to predict riming without the use of MDV. The predictions of both ANNs for a wintertime convective cloud fit coinciding in-situ observations extremely well, suggesting the possibility to predict riming even within convective systems. Application of ANN #2 to an orographic case yields high $\text{FR}_{ANN}$ values coinciding with observations of solid graupel particles at the ground.

## 1 Introduction

Mixed-phase clouds are important components of the climate system, because they play a major role both for the radiation budget (Tan et al., 2016) and for the hydrological cycle (Mülmenstädt et al., 2015). In this type of cloud, ice particles and supercooled liquid water (SLW) can coexist in the temperature range between 0 and $-38°C$. The availability of both ice and SLW allows for riming, i.e. the accretion and freezing of SLW on frozen ice particles. Riming is a key process for ice growth

and eventually precipitation formation (Lamb and Verlinde, 2011; Heymsfield et al., 2020). Detecting and quantifying riming using ground-based remote sensing observations is a non-trivial task, but of great interest for several reasons:

Firstly, ice-phase microphysical growth process rates in general are associated with a large uncertainty, posing a major challenge for microphysics schemes in numerical weather forecast and climate models (Morrison et al., 2020). Recently, considerable effort has been made to improve the representation of riming in models, ranging from process-oriented approaches (e.g. Seifert et al., 2019) to novel microphysics schemes (e.g. Morrison and Milbrandt, 2015). Deriving the rime mass fraction from remote sensing measurements would enable observation-model comparison studies, which are an important step to evaluate and constrain parameterizations of riming in models. Secondly, those regions where riming occurs in clouds coincide with icing conditions, which pose a hazard to aircraft traffic (Cao et al., 2018). The ability to detect riming conditions e.g. using ground-based vertically pointing remote sensing instruments would allow for real-time warnings near airports (Serke et al., 2010). Finally, long-term statistics of riming are needed in order to better understand under which conditions riming is taking place in the atmosphere.

In the past, several approaches to retrieve riming from ground-based remote sensing observations have been developed. Riming, at least in its initial stage, increases particle density but not much its size, because the freezing SLW droplets first fill the cavities in the particle structure (Heymsfield, 1982; Seifert et al., 2019). As an effect, its terminal velocity increases (Mosimann et al., 1994; Mosimann, 1995), and also the aspect ratio and backscattering cross-section are changed (Garrett et al., 2015; Leinonen and Szyrmer, 2015). While it is quite challenging to detect riming in polarimetric measurements (Vogel and Fabry, 2018; Li et al., 2018), robust signatures were identified by putting the radar signals of multiple frequencies in relation: Kneifel et al. (2015, 2016) utilized observations of collocated cloud radars with three different wavelengths to pin down fingerprints of riming and aggregation in the triple-frequency Doppler spectral space. They found that rimed particles can be connected with a combination of low dual-wavelength ratios (DWR) of X and Ka-band radar ($DWR_{X,Ka} < 3$ dB) and $DWR_{Ka,W}$ values larger than 3 dB. These signatures are clearer for larger particles. For smaller particles, i.e. small DWR values, this distinction becomes ambiguous. Under these conditions, the fall velocity of the hydrometeors can give an additional constraint. Mason et al. (2018) developed an optimal estimation retrieval to obtain a density factor parameter, which is sensitive to riming, using observations of mean Doppler velocity (MDV) and $DWR_{Ka,W}$. Li et al. (2020) developed a snow observation classification with a rimed and an unrimed category, using MDV and $DWR_{X,Ka}$. Oue et al. (2021) combined MDV and $DWR_{Ka,W}$ with polarimetric observations and were able to observe even different stages of the riming process. However, only a few sites worldwide are equipped with cloud radars of two or more different frequencies, and correct alignment and volume matching is associated with a certain effort. Another approach which does not rely on observations at multiple wavelengths is based on MDV only (Mosimann et al., 1994; Mosimann, 1995). Recently, this technique has been further developed by Kneifel and Moisseev (2020), who were able to derive a robust estimate of rime mass fraction using MDV averaged in both spatial and temporal domains over a 100 m / 20 min height-time window. They applied this method to time series of radar measurements at several stations located within the CloudNet (Illingworth et al., 2007) network.

While there is an evident correlation between MDV and riming due to the larger density and thus increased fall speed of rimed ice particles, it is not always possible to rely on this relation. The method fails when the assumption that the MDV is equivalent

to the particle fall speed in the observation volume does not hold. This is, e.g., the case in convective systems, which can cause persistent up- or downdrafts. Also, in complex terrain, orographically induced waves can shift the observed MDV up or down by several meters per second. These gravity waves can be temporally persistent and consequently not be removed by temporal averaging (Radenz et al., 2021), making MDV-based approaches such as the one by Kneifel and Moisseev (2020) unfeasible.

This is unfortunate especially considering the fact that riming plays an important role in the microphysics of convective and orographic systems (Woods et al., 2005; Houze and Medina, 2005). In these cases, other radar variables have to be exploited in search for fingerprints of riming, which allow for a quantitative detection of this process. During riming, multiple hydrometeor populations are present in the same radar observation volume. Due to their different terminal fall velocities, these hydrometeor types result in multiple peaks in the cloud radar Doppler spectra (Kalesse et al., 2016, 2019). A broadening of the spectra is the

result, which is e.g. evident in an increase of the spectrum width (the second moment). Another variable which is impacted by riming is the skewness (the fourth moment): Multi-peaked spectra have nonzero abolute skewness values, zero being the value of a Gaussian distribution. The impact of riming on these other radar variables is, however, not as straightforward as for MDV. For example, riming and aggregation processes may result in similar signatures, because aggregation can also lead to bimodal spectra (e.g. Barrett et al., 2019). Consequently, the often ambiguous information contained in the higher radar moments cannot

easily be extracted from the radar variable space using e.g. simple thresholding techniques (Maahn and Löhnert, 2017). More sophisticated methods to derive the relationship between riming and the set of available radar variables are required.

Machine learning (ML) algorithms offer ways to discern relationships from complex data sets. The interest in ML techniques has been increasing rapidly over the past years, and exciting advances have been accomplished in many scientific fields. Also in atmospheric sciences, the use of ML offers a promising path for scientific discoveries (e.g. Seifert and Rasp, 2020). Deep

learning is a type of ML, where artificial neural networks (ANNs) are trained to make predictions. Their potential to extract useful relations from ground-based radar observations has been demonstrated in work published e.g. by Luke et al. (2010) and van den Heuvel et al. (2020).

In this study, the overall goal is to develop a technique which does not rely on MDV and can consequently be applied to data sets acquired in complex terrain, where orographically induced vertical air motions render MDV unusable. More precisely,

our motivation is to derive riming estimates for a data set acquired in Punta Arenas, Chile. This site is strongly influenced by stationary gravity waves, which make the application of MDV-based riming retrievals challenging. At the same time, information about the occurrence of riming over Punta Arenas would be of special interest due to its location in the vicinity of the Southern Ocean and the pristine aerosol conditions encountered there (Foth et al., 2019; Radenz et al., 2021). We assess in this work how ANNs can be used to predict riming using ground-based zenith-pointing cloud radar measurements as input

features. We optimize and train ensembles of ANNs using different combinations of radar variables. These variables include the equivalent radar reflectivity factor (Ze), the MDV, the spectrum width from left to right edge of the spectrum above the noise floor ("spectrum edge width", SEW), and the skewness.

## 2 Data and methods

### 2.1 Field experiments

We are using data from the "Biogenic Aerosols – Effects on Clouds and Climate" (BAECC; Petäjä et al., 2016) campaign to train, validate and test ensembles of ANNs. During BAECC, in situ observations of snow were collocated with Ka- and W-band radar measurements at a site in Hyytiälä, Finland. The ensembles of trained ANNS are then applied to observations collected during the "TRIple-frequency and Polarimetric radar Experiment for improving process observation of winter precipitation" (TRIPEx-pol; Mróz et al., 2020) in Section 3.2. The motivation to use these data is twofold: Firstly, it allows us to demon-

strate that the ANNs are able to generalize to different meteorological conditions and radar settings, and to check whether the predictions are consistent between the Ka- and W-band setup. Secondly, the riming retrieval can be additionally compared with expected signatures of riming in triple-frequency radar observations. Another validation study is performed using data obtained at the Leipzig Institute for Meteorology (LIM) on 19 March 2021 in Section 3.3. In this data set, W-band radar data are complemented by ground-based in situ observations of graupel and snowflakes. In Section 3.4, we apply one set of ANNs

to a graupel case measured by the same W-band radar during the "Dynamics, Aerosol, Cloud And Precipitation Observations" (DACAPO-PESO) field experiment in Punta Arenas. In this section, each of the four measurement setups is briefly introduced.

### 2.1.1 The BAECC experiment

The BAECC campaign was a joint effort between the US Department of Energy Atmospheric Radiation Measurement Pro-

gram (DOE ARM), University of Helsinki, the Finnish Meteorological Institute (FMI), the US National Aeronautics and Space Administration (NASA), and Colorado State University. From February to September 2014, the second ARM mobile facility (AMF2) was deployed at the Station for Measuring Ecosystem-Atmosphere Relations II (SMEAR II) in Hyytiälä, Finland (61° 51′N, 24° 17′E). A detailed description of the setup of the remote sensing and in situ instrumentation can be found in Petäjä et al. (2016). The suite of remote-sensing instruments include a Ka-band ARM Zenith-pointing radar (KAZR), and a

Marine W-Band ARM Cloud Radar (MWACR), both Doppler cloud radars. Moreover, several ground-based in situ instruments for measuring microphysical properties of snow were deployed at the site. We use data from the ground-based in situ Precipitation Imaging Package (PIP, Pettersen et al., 2020), a video disdrometer which measures the size and velocity of hydrometeors reaching the surface.

### 2.1.2 The TRIPEx-pol experiment

The TRIPEx-pol campaign took place from November 2018 until February 2019 at the Jülich Observatory for Cloud Evolution Core Facility, Germany (JOYCE-CF, 50°54′N, 6°25′E, Löhnert et al. (2015)). At that site, vertically pointing pulsed X-, and Ka-band (Görsdorf et al., 2015) systems (manufactured by Metek GmbH) and a Frequency Modulated Continuous Wave

vertically-pointing W-band radar (FMCW, Radiometer Physics GmbH, Küchler et al., 2017) are employed on a roof platform

within 10 m distance. Calibration, quality control, matching and resampling procedures described in Dias Neto et al. (2019) were applied to yield high-quality data from all three instruments on the same time-height grid. We are using data from seven days between 24 November 2018 and 10 January 2019 during which thick cloud systems were observed, covering a range of meteorological conditions. The specific days chosen for this analysis are 24 November, 3, 8, 23, and 27 December, and 7 and 10 January, all featuring thick mixed-phase cloud cases, possibly with riming.

### 2.1.3 LIM roof platform

The University of Leipzig Institute for Meteorology (LIM) remote sensing instrument suite is located on the observatory's roof platform (51°20′N, 12°22′E). This CloudNet site encompasses LIM's 94 GHz Radar (LIMRAD94), and a Video In Situ Snowfall Sensor (VISSS, Maahn et al., 2021). LIMRAD94 is a W-band FMCW radar very similar to the system which was employed during TRIPEx-pol. The VISSS is an optical observation system for snow particles, measuring 140 frames per

second at an optical resolution of 59 $\mu$m. While particle size distribution and fall velocity are planned to be avaliable as data products in the future, we here make use of the recorded images of snow particles in a qualitative manner.

### 2.1.4 The DACAPO-PESO experiment

Since December 2018, the Leipzig Aerosol and Cloud Remote Observations System (LACROS, Bühl et al., 2016) has been employed in Punta Arenas, Chile (53°10′S, 70°56′W). LACROS comprises a suite of active and passive remote sensing in-

struments, and for the time period from December 2018 to October 2019, LIMRAD94 was employed next to the LACROS shipping container. The measurement site in Punta Arenas is located at the most southerly continental part of South America, where the meteorological conditions are characterized by prevailing strong westerly to northwesterly winds. As the air flow hits the continental mass, it is forced over mountainous orography and at the same time strongly decelerated, resulting in orographic wave motions. From the ground-based radar perspective, persistent up- or downdrafts (lee waves), as well as updrafts followed

by downdrafts or vice versa can often be observed. The site and the DACAPO-PESO experiment (http://dacapo.tropos.de) have been described in more detail in Floutsi et al. (2021).

### 2.1.5 Attenuation corrections

Radar reflectivity of all data used in this study were corrected for attenuation by atmospheric gases, liquid water, melting

particles and ice. The Passive and Active Microwave TRAnsfer (PAMTRA) forward operator (Mech et al., 2020) was used in combination with CloudNet products, which are either freely available on the CloudNet data portal (https://cloudnet.fmi.fi/) or were processed locally using the cloudnetpy Python package (Tukiainen et al., 2020): Attenuation due to atmospheric gases, including water vapor, was estimated using PAMTRA and the profiles of temperature and humidity included in the CloudNet model files. For liquid water, we used the measured liquid water path and distributed the mass among the pixels classified as

"liquid containing" in the CloudNet classification mask, weighted by the measured reflectivity. We used PAMTRA to obtain the attenuation caused by the mass liquid in the respective height bin, assuming a monodisperse particle size distribution for cloud droplets and an exponential distribution for rain (Mech et al., 2020). This means that attenuation caused by SLW droplets is only corrected for in the SLW layers detected by the CloudNet classification mask, i.e. limited by lidar signal attenuation. If pixels were classified as "melting" in CloudNet, the melting layer attenuation was assumed following Matrosov (2008), who derived relations for Ka- and W-band as a function of rainfall rate. Attenuation due to snow and ice was neglected for Ka-band and estimated according to Protat et al. (2019) for the W-band radars. If the cumulative attenuation correction for a pixel exceeded 10 dBZ, the profile was removed from the analysis.

## 2.2 Sampling of training data

Masses of individual snow flakes can be retrieved by applying hydrodynamic theory to PIP observations of particle velocity and size (von Lerber et al., 2017). A data set containing observed snow particle number size distributions, along with retrieved particle masses collected between 2014 and 2015 in 5-minute temporal resolution is freely available on GitHub (Moisseev, 2018). Using these retrieved masses $m$ and the maximum dimensions of the observed particles, $D_{max}$, we first estimated the rime fraction. The mass of unrimed snow, $m_{us}$ is assumed as in Moisseev et al. (2017); Li et al. (2020):

$$m_{us} = \alpha \cdot D_{max}^{\beta} \tag{1}$$

where the values for $\alpha$ and $\beta$ are $\alpha = 0.0053$ and $\beta = 2.05$ (in cgs units). The rime fraction is then defined as in Kneifel and Moisseev (2020):

$$FR_{PIP} = 1 - \frac{IWC_{us}}{IWC} = 1 - \frac{\int m_{us}(D_{max}) \cdot N(D_{max}) \mathrm{d}D_{max}}{\int m(D_{max}) \cdot N(D_{max}) \mathrm{d}D_{max}} \tag{2}$$

Here, $IWC_{us}$ is the estimated ice water content for unrimed snow having the same N($D_{max}$) as the observed particles. It is obtained by integrating $m_{us}$ computed from Eq. (1) over the measured particle number size distribution $N(D_{max})$. FR$_{PIP}$ values smaller than zero were removed. In the next step, continuous PIP sampling periods longer than one hour were identified for which KAZR and/ or MWACR observations are available as well. This results in six snow cases between 1 February and 20 March 2014, which are listed in Table 1.

During BAECC, a turbulent surface layer was often present, resulting in a broadening of Doppler spectra and impacting their width and other higher-order moments relevant for the training. The following sampling procedure for the training data set was chosen, striving to achieve the best-possible spatio-temporal match between remote sensing and in situ observations, while at the same time avoiding to sample spectra which are strongly impacted by surface induced turbulence: We compute the turbulent eddy dissipation rate (EDR) for 5-minute time periods (Maahn et al., 2015) and determine $r$, the lowest range at which EDR is $< 10^{-3} m^2 s^{-3}$. This threshold was determined empirically in a sensitivity study, in which spectral broadening was simulated using convolution of a spectral broadening term $\sigma_T$ with measured Doppler spectra. The time $\Delta t$ it takes for the particle to travel from height $r$ to the surface can be estimated using the measured MDV at $r$, making the assumption of temporal and spatial coherence, i.e. that the properties measured at time $t$ and $r$ will be the same as the properties at time $t + \Delta t$

**Table 1.** Overview of the cases used in the training set, and coincident notable events according to Table 2 in Petäjä et al. (2016)

| time period | precipitation type | note |
|---|---|---|
| 01 February 2014, 01:07 – 01 February 2014, 05:57 UTC | snow | |
| 01 February 2014, 09:07 – 01 February 2014, 15:57 UTC | snow (riming) | |
| 15 February 2014, 20:07 – 16 February 2014, 01:57 UTC | snow (riming) | |
| 21 February 2014, 15:07 – 22 February 2014, 03:27 UTC | snow (riming)/ melting snow | |
| 18 March 2014, 23:07 – 19 March 2014, 19:57 UTC | large aggregates/ riming | no MWACR data |
| 20 March 2014, 15:07 – 20 March 2014, 23:47 UTC | snow/ riming | |

at $r + \Delta t \cdot \mathrm{MDV}(t,r)$. Note that negative MDV values indicate downward motions. A schematic for visualization of the applied sampling technique is shown in Fig. 1. For each PIP measurement, the rime fraction was obtained using Eq. 2. All the KAZR and MWACR spectra at range $r$, which were matched with a 5-minute PIP measurement were extracted, and Ze, MDV, and skewness were computed. MDV was corrected for air density change as a function of altitude (Vogel and Fabry, 2018). An additional measure of the width of the spectra was computed and stored: The SEW is defined as the distance (in $\mathrm{m\,s^{-1}}$) between the left edge and the right edge above the noise threshold. This threshold is the mean noise according to Hildebrand and Sekhon (1974) plus three standard deviations of the noise. SEW offers the advantage that it is, in contrast to the spectrum width, not weighted by the reflectivity, making it more sensitive to broadening of the spectrum due to small peaks, e.g. caused by liquid water. While SEW and skewness are not among the "traditional" radar moments, they are often available; e.g., the ARM MicroARSCL product contains the skewness and edges of the spectra (Kollias et al., 2007; Luke et al., 2008).

The resulting extracted training data set includes 105,115 Ka-band and 209,965 W-band radar observations along with the $\mathrm{FR}_{PIP}$ retrieved from the corresponding PIP measurements. It is available on GitHub (https://github.com/ti-vo/BAECC_features). A 3D plot of three of the contained features (Ze, SEW and skewness) colored by $\mathrm{FR}_{PIP}$ is shown in Fig. 1c. The video supplement contains an animated visualization of the same variables.

## 2.3 Machine learning methods

The retrieval described in this section reflects a regression problem. Machine learning techniques are used to relate Doppler cloud radar moments and SEW to $\mathrm{FR}_{PIP}$ using a fully connected neural network. Multiple ANN models are trained to make predictions, given many input (features) and output (target) pairs, with the goal to search for a function that both fits the given data well, and also is able to generalize to new values (Goodfellow et al., 2016). For the steps described in the following, tools provided in the Python library *Sklearn* (Pedregosa et al., 2011) were used.

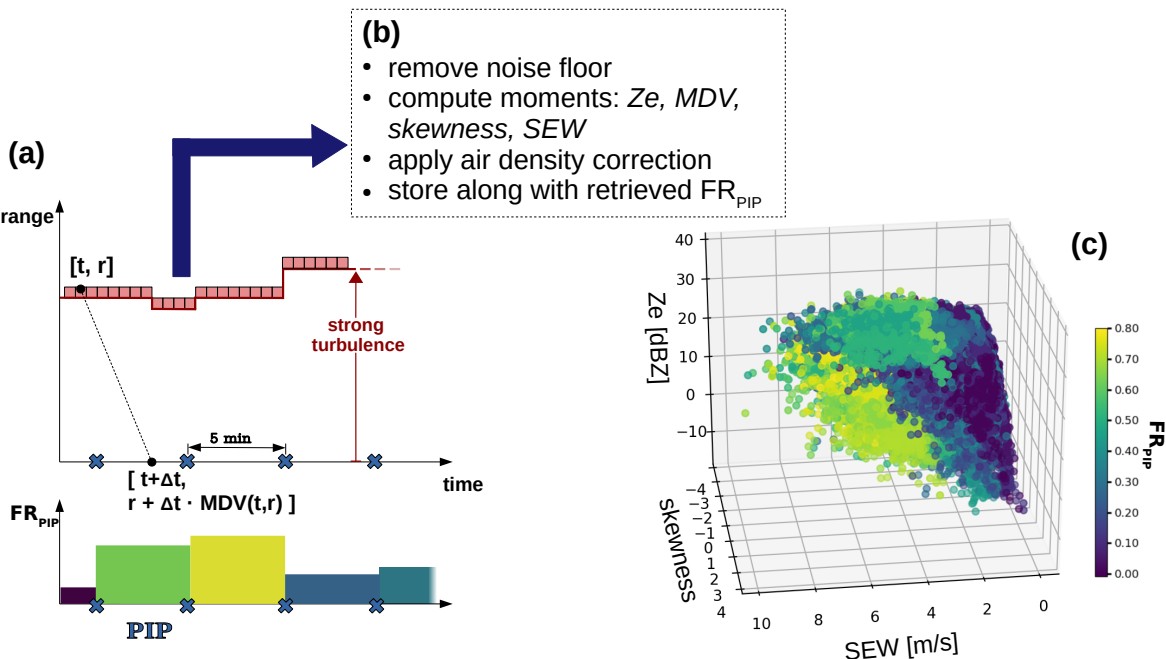

**Figure 1.** (a) Schematic of the spatio-temporal matching of the in situ (PIP) observations and the cloud radar observations: Radar spectra at time $t$ and range $r$ are assigned to the PIP measurement closest to the time $t+x$. (b) Processing steps applied to each extracted radar Doppler spectrum. (c) 3D plot of three features contained the resulting training data set (equivalent radar reflectivity factor Ze, spectrum edge width SEW, and skewness) colored by the rime mass fraction FR.

### 2.3.1 Data preparation

Radar moments and the SEW extracted from the BAECC data set are used as input features, and $FR_{PIP}$ values are the desired target. One problem emerges because only a limited number of cases with high FR, i.e. values $> 0.7$ were observed during the period when KAZR, MWACR and PIP were collocated in Hyytiälä. By their nature, high FR cases are rather short-lived, so the duration and therefore length of such observations are rather small. This skewed distribution of target values has impacts on the training of machine learning algorithms and needs to be taken into account carefully. ANN models will primarily be trained to predict values in the range where most of the observations lie. In order to ensure that trained models also succeed to predict high values of FR, which are of special interest, the training data set is resampled using the synthetic minority oversampling technique for regression problems (SMOTER, Torgo et al., 2013). SMOTER is an algorithm which oversamples rare cases and undersamples frequent cases in the training data set, leading to a more balanced distribution.

For many ML applications, standardization of the input data set is a common requirement. In order to avoid effects caused by outliers, it is advantageous to use a robust scaler, which scales each of the input features according to the inter-quartile range (IQR) and removes the median. The same robust scaler which is defined using the training data set will also be applied later on

to new data being sent through the model.

After scaling, the labeled data are split into two parts, one training and validation set (90%), and retaining 10% of the data for the testing phase. The choice of these ratios is subjective and associated with a tradeoff between learning and the assessment of the generalization ability.

### 2.3.2 ML model specifications

In our study, we are using a multilayer perceptron (MLP), which is composed of one input layer, at least one hidden layer, and one output layer. In this section, the hyperparameters, which determine the network's architecture and the training process, are discussed.

The input layer consists of one neuron for each input feature, and the output layer yields the output value(s). In between the input and output layers, one or more hidden layers are defined. Each hidden layer contains a certain number of neurons, which transform the values from the previous layer and then apply a non-linear (rectified linear unit; $f(x) = max(0, x)$) activation function. In a fully connected model, a given neuron is connected to every neuron in the previous layer. During training, an optimizer iteratively adjusts the weights of the model, until a minimum in error ("loss") is reached. This error is defined by the loss function, in our case the squared error. We are using the "Adaptive Moment Estimation" (Adam) optimizer, a stochastic gradient-based method (Kingma and Ba, 2017), which requires computing the gradient of the loss function with respect to the model parameters by a back-propagation algorithm. The learning rate controls the step-size for updating the weights, and is in our case set to a constant value of 0.001.

The number of hidden layers (which is also referred to as the layer depth) and respective number of neurons are hyperparameters, which require tuning in the validation phase. In Goodfellow et al. (2014), plots of accuracy vs.layer depth are used to determine the optimal ANN architecture. Following this concept, we plotted the root mean squared error (RMSE) of the validation data set predictions and targets vs. the number of neurons/ hidden layers (not shown) to find an architecture which is simple yet accurate. For the limited number of input features used in our approach, a relatively simple setup with one hidden layer and two neurons fulfills these requirements.

Here, we use a k-fold cross validation (k-fold cv) approach with three folds. This means that the training and validation set is split into three equally-sized chunks (folds). Each of the folds is once used to test the model while the remaining two folds are used to train the model. This method enables us to compute the mean error of the three folds, which is a more reliable measure of the ANN performance than the error obtained for one split between training/ validation set only. In addition, this approach results in three trained models, which can be applied as an ensemble to yield an average prediction and a spread, which can (at least to some degree) be regarded as an uncertainty estimate of the prediction. In this study, we are using three different parameter combinations as input features to the ANNs:

1. ANN #0: Ze, MDV

2. ANN #1: Ze, MDV, SEW and skewness

3. ANN #2: Ze, SEW and skewness

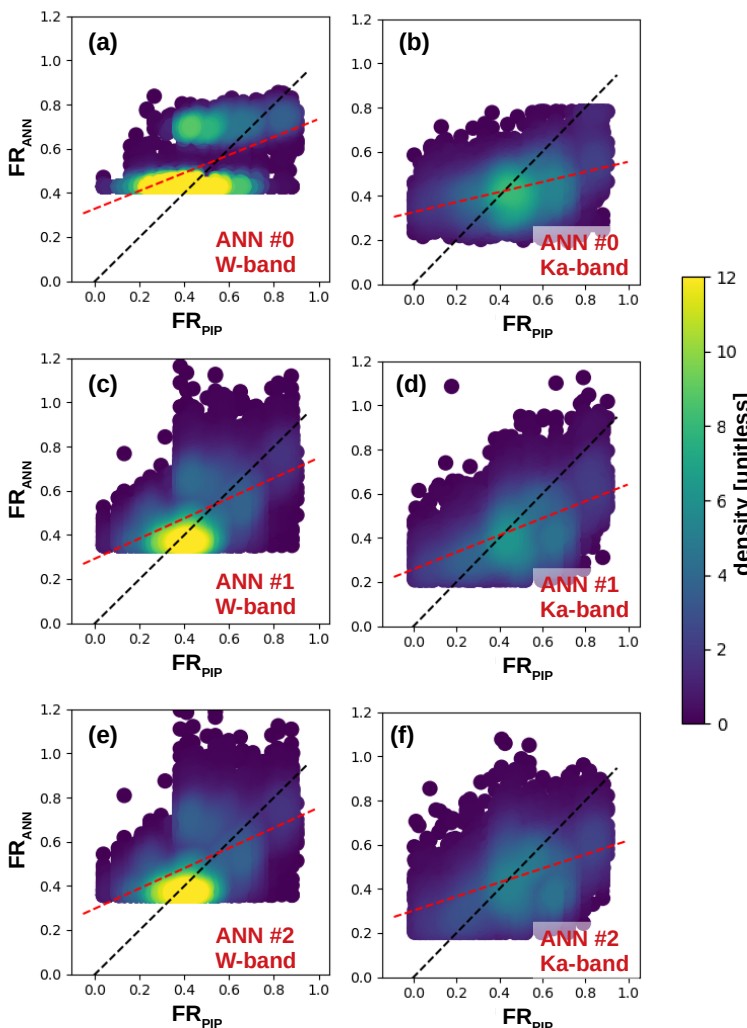

**Figure 2.** Scatter plots of ANN performances on the test set. The black dashed line marks the 1:1 line, the red dashed line is a linear model fit to the data. The slopes of the red dashed lines are listed in Table 2. (a) W-band, ANN #0; (b) Ka-band, ANN #0 (c) W-band, ANN #1; (d) Ka-band, ANN #1; (e) W-band, ANN #2; (e) Ka-band, ANN #2

The set of input parameters in ANN #0 was chosen because they are similar to existing riming retrieval methods relying on MDV. In order to check whether adding more radar variables can improve the riming estimate, ANN #1 uses the SEW and skewness as additional input features. ANN #2 represents the set of input parameters if MDV can not be used e.g. due to persistent up- or downdrafts.

In the testing phase, the model performance is evaluated using the test set prediction RMSE, in connection with visual in-

**Table 2.** Test set performances for the three different ANN ensembles, for the complete test set, and a subset of it containing only high ($> 0.5$) target values

| ANN | test set RMSE Ka-band | test set RMSE Ka-band($FR > 0.5$) | test set RMSE W-band | test set RMSE W-band($FR > 0.5$) | slope Fig. 2 Ka-band | slope Fig. 2 W-band |
|------|------|------|------|------|------|------|
| ANN #0 | 0.21 | 0.23 | 0.15 | 0.15 | 0.23 | 0.41 |
| ANN #1 | 0.18 | 0.19 | 0.15 | 0.14 | 0.38 | 0.46 |
| ANN #2 | 0.20 | 0.20 | 0.15 | 0.15 | 0.32 | 0.46 |

spection of scatter plots of ANN test set predictions and target values. In addition, time-height plots of ANN predictions are examined with regard to physical plausibility of the predicted features. We decided to put an additional focus on the ability of the networks to predict high FR values $\geq 0.5$. The reason for this choice is that the extracted Doppler spectra features are expected to contain little information about riming up to a moderate riming stage. For high FR values, the prediction should be more accurate because the riming signal should be clearer in the input features. Table 2 summarizes the test set RMSEs found for the three different ANN input parameter sets and the two radar frequencies. We will refer to the predicted quantity in the following as $FR_{ANN}$, to distinguish it from the $FR_{PIP}$ retrieved from in-situ observations. Fig. 2 shows scatter plots of ANN predictions and test set $FR_{PIP}$ target values, for the three different ANN ensembles, for Ka- and W-band. The slopes of the linear models (red dashed lines in Fig. 2) are listed in Table 2. From Fig. 2 it becomes apparent that ANN #0 seems to have a problem in the W-band set up, because two populations of values are separated in the predictions. The slopes of the linear fits, which should ideally be 1, are lowest for ANN #0 for both considered radar frequencies. This observation is however not reflected in the RMSEs listed in Table 2, which hardly differ between the three ANN ensembles for the W-band. The RMSEs in turn show that ANN #0 performs worse than the other two setups for the Ka-band. This illustrates the limits of using a single quantity like RMSE as a quality metric. ANN #1 and #2 also have issues predicting high FR values, and all ANNs feature a cut-off at low FR values. This might be because riming signatures for low FR are not very pronounced in the cloud radar Doppler spectra, making it impossible for the ANNs to accurately predict $FR \leq 0.3$ from the extracted features. Similarly, the issues of the ANNs to predict high FR values could be explained by spectra of pure graupel not necessarily being bimodal, which would result in a less clear signature in SEW and skewness. For the remaining study, we will focus only on ANN #1 and ANN #2.

## 2.4 Error consideration

We acknowledge that several sources of uncertainty impact the riming predictions, only some of which can be quantified. For the training data set, assumptions about the density of unrimed snow and the viewing geometry corrected $D_{max}$ were made. The m-D relation used to derive the mass of unrimed snow (Moisseev et al., 2017) is assumed to overestimate $m_{us}$ by a maximum of 5%. Furthermore, errors are introduced by the spatio-temporal matching of radar and PIP observations. To get

a grasp of the overall effect of measurement uncertainties, which propagate into the $FR_{ANN}$ predictions, a sensitivity study was performed. We assumed an error of 0.2 dBZ for Ze, one Doppler bin for MDV and SEW, and 0.2 for the skewness, and added this uncertainty to selected Doppler spectra. This was accomplished by random picking of n = 10,000 samples from a Gaussian distribution defined by the measured values as mean and above errors as standard deviation. The resulting standard deviation in the predicted $FR_{ANN}$ is approximately 1%. In addition to the 5% uncertainty from the mass of unrimed snow, and other error sources which cannot be quantified we propose to assume an overall uncertainty of approximately 10% due to the training data. This uncertainty has to be added to the uncertainty of the ANN models (Table 2).

## 3   Results and Discussion

This section is structured as follows: In Section 3.1, a well-defined "benchmark" riming case from the BAECC dataset is presented. The application of the ANNs to the TRIPEx-pol dataset is demonstrated in Section 3.2. In Section 3.3, the performance of the ANNs for the W-band radar in Leipzig on 19 March 2021 is evaluated by comparison to in situ observations. Finally, in Section 3.4, we present ANN predictions for a case obtained during the DACAPO-PESO field campaign using the same W-band radar.

### 3.1   BAECC benchmark riming case

In addition to using the BAECC data set for training, validation and testing, we also capitalize on the fact that these data have already been extensively studied with respect to microphysical growth processes, and several well-defined case studies are available. We evaluate the performance of our newly developed riming estimation technique for a 1-hour period between 22:00 and 23:15 UTC on 21 February 2014, which is part of the 10% of the labeled data that were retained for the test set. Riming was taking place starting at around 17:00, the LWP reaching its maximum value of more than 1000 g m$^{-2}$ around 22:00 UTC (Fig. 4e, cf. Moisseev et al., 2017). Parts of the focus period considered here between 22:00 and 23:15 UTC have been previously analyzed in detail e.g. by Kalesse et al. (2016, 2019), Moisseev et al. (2017), Mason et al. (2019) and Kneifel et al. (2015, 2016). Fig. 3 shows Ze, MDV, spectral width, skewness and SEW measured by the KAZR. The case is characterized by a "seeder-feeder" situation, where a frontal snow cloud merges into a mid-level mixed-phase cloud. In the mixed-phase cloud, SLW layers are present at 0.7 to 0.9 km, and slightly below 3 km. As snow starts to fall from the frontal ("seeder") cloud into the lower-level ("feeder") cloud, intense riming happens along a slanted fall-streak feature at around 22:40 to 22:45 UTC, resulting in Doppler spectra with multiple peaks (Kalesse et al., 2016, 2019). During the following time period, a transition from strongly rimed particles to unrimed snow aggregates at between 23:03 and 23:10 UTC was observed by Kneifel et al. (2015, 2016); Moisseev et al. (2017); Mason et al. (2019).

Fig. 4 shows the predicted $FR_{ANN}$ for the two different ANN ensembles, and the two different radar frequencies, respectively. The ensembles yield very similar predictions, which is remarkable given the fact that ANN #2 does not use MDV as input feature. The surface-induced turbulent layer close to the ground is masked with grey pixels, where EDR exceeds the threshold of $10^{-3}$ m$^2$s$^{-3}$. For the W-Band, columns where the estimated attenuation exceeded 10 dBZ, were masked. For all four

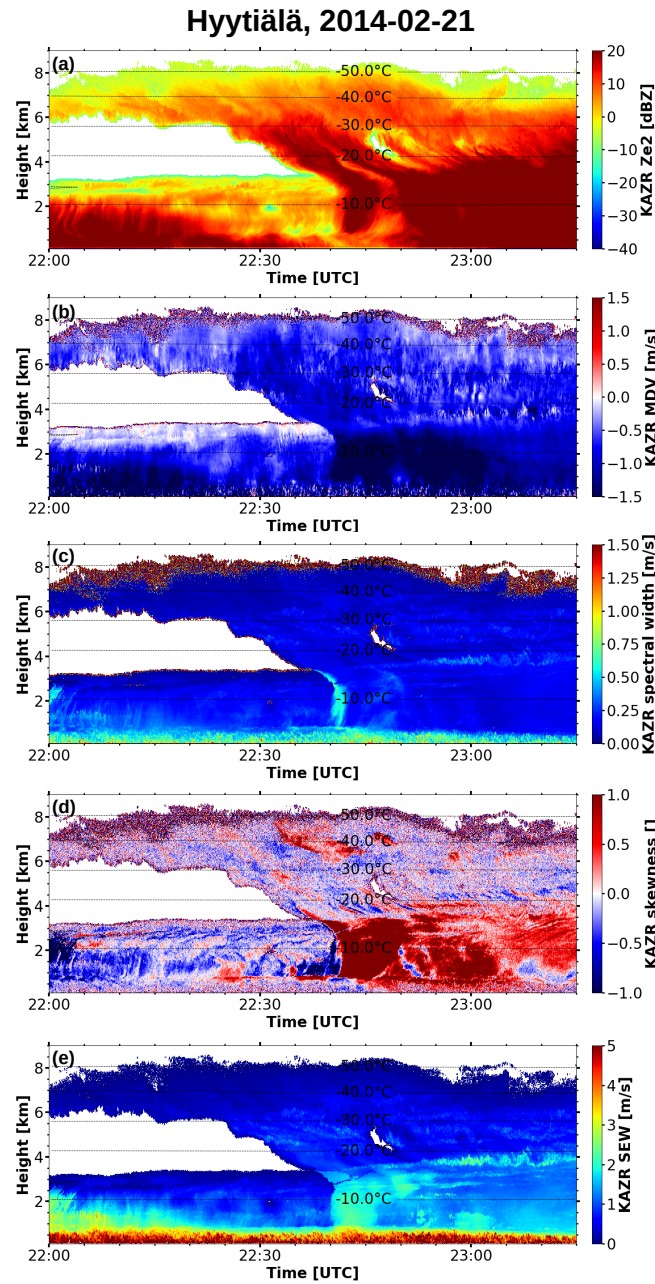

**Figure 3.** Radar moments measured by the KAZR in the focus period between 22:00 UTC and 23:15 UTC on 21 February 2014. a) Equivalent radar reflectivity of the full spectrum in dBZ; b) mean Doppler velocity (negative values indicating downward motion) computed from the full spectrum; c) spectral width computed from the full spectrum; d) skewness computed from the full spectrum; e) spectrum edge width

**Hyytiälä, 2014-02-21**

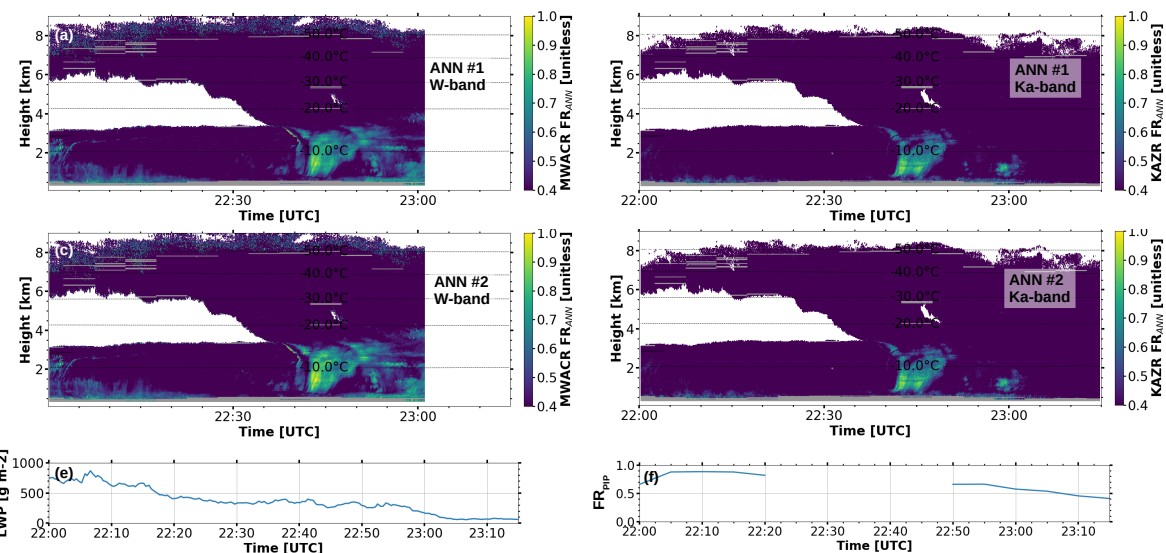

**Figure 4.** Riming during the focus case on 21 February 2014 predicted by (a) and (b) ANN # 1 trained on MWACR and KAZR data, respectively (Ze, MDV, SEW and skewness); (c) and (d) ANN # 2 trained on MWACR and KAZR data, respectively (Ze, SEW, and skewness); (e) Liquid water path obtained from microwave radiometer measurements; (f) $FR_{PIP}$ measured during the event. Pixels with EDR above the threshold of $10^{-3} m^2 s^{-3}$ are masked in grey in (a)-(d).

configurations, a clear increase in $FR_{ANN}$ is visible at around 22:40 UTC, when snow starts falling from the seeder cloud through the SLW layers in the lower-level mixed-phase (feeder) cloud. This is the period for which Kalesse et al. (2016, 2019) reported riming signatures in Doppler spectra featuring multiple peaks. Unfortunately, due to low precipitation intensity at the
5 ground, no PIP-based FR retrieval was avaialble for this time period (Fig. 4f). However, a continuous decrease in LWP points to the depletion of SLW by the strong riming (Fig. 4c). Later, after around 22:50 UTC, the predictions differ between Ka- and W-band: While the two ANN ensembles trained on Ka-band data only predict riming during a short period around 23:00 UTC, the $FR_{ANN}$ predictions in the W-band setup remain elevated up to a range of around 4 km. The Ka-band ANNs are apparently able to detect the transition from increased riming to less riming around 23:05 UTC, which was reported in existing studies of
10 the event. Around this time, the LWP has reached its minimum, indicating that no SLW for further riming is available in the column. Coinciding with this period from around 23:05 UTC onwards, the $FR_{ANN}$ is very low ($\leq 0.5$) throughout the cloud system, which is in accordance with the measured decrease of $FR_{PIP}$. (Fig. 4f). This first case study is promising with respect to the usability of our trained ANNs to predict riming from cloud radar moments. Furthermore, the similarity between ANN#1 and ANN #2 for both radar frequencies shows that that their application might be possible even without the use of MDV.

 **3.2 TRIPEx-pol case study and triple-frequency signatures for seven cases**

To answer the question whether the developed methods are able to generalize to conditions different than the ones they were trained on, considering another data set is required. For this reason, the two ANN sets are applied to data from a different site, obtained by radars with different settings. We will first focus in more detail on the 24 November 2018 case from the TRIPEx-pol campaign. This precipitation event has been analyzed with respect to rain and ice microphysics by Mróz et al. (2020), who

found strong signatures of aggregation during the period from 06:45 to 07:45 UTC, and riming during a shorter time interval around 09:00 UTC.

Fig. 5 shows the radar moments measured during the period from 02:00 UTC to 11:30 UTC on 24 November 2018. The period characterized as "aggregation" by Mróz et al. (2020) clearly shows up as a patch of increased signal in the $DWR_{X,Ka}$ between 1 and 4 km range, while during the "riming" period around 09:00 UTC, an obvious increase in absolute MDV values can be

observed in a similar range interval, along with an increase in SEW. In Fig. 6, the predictions of the two ANNs are shown side by side for W-band and Ka-band. In both cases, the CloudNet classification mask was used to only apply the ANNs to those parts of the cloud which were classified as ice or ice and liquid. For the two different frequencies, as well as for the two ANNs, the predicted $FR_{ANN}$ features are very similar. For the "aggregation" period, which has the strongest signal in the $DWR_{X,Ka}$ (Fig. 5c) from approximately 06:45 to 07:30 UTC, some riming is predicted, however relatively low values. During

the "riming" period, which clearly shows up as increased MDV in Fig. 5b between 08:00 and 09:00 UTC, strong riming is predicted for both wavelengths, and by both ANN sets. Again, it is remarkable how the MDV features in Fig 5b can be discovered even in the predictions by ANN #2, which does not use MDV (two lower panels in Fig. 6). Despite the differences in scattering properties and attenuation due to hydrometeors at Ka- and W-band as well as the different noise levels of the two radars, the retrieved $FR_{ANN}$ is rather similar. The common time-height grid in the TRIPEx-pol data set allows for convenient

direct comparison of the two radar frequencies for each of the ANN set ups: The correlation of the predicted $FR_{ANN}$ values for all considered TRIPEx-pol cases is high for both ANN sets, the $R^2$ of the Ka- vs. W-band-based predictions being 0.73 for ANN #1 and 0.81 for ANN #2, respectively.

As mentioned earlier on, riming and aggregation can produce distinct signatures in the triple-frequency space of X-, Ka- and W-band reflectivity. When considering a plot with $DWR_{X,Ka}$ on the abscissa and $DWR_{Ka,W}$ on the ordinate, observations

obtained during riming events fall onto a line at low $DWR_{X,Ka}$ (pink line in Fig. 7). Aggregates, in contrast, tend to yield a hook-like feature, due to their comparably large size at which $DWR_{Ka,W}$ is in saturation because of Mie scattering. This conceptual model was presented by Kneifel et al. (2015) and further explored by Mason et al. (2019), who found that not only the density, but also the shape parameter of the particle size distribution and the internal structure of aggregates can have strong impacts on triple-frequency signatures.

In Fig. 7, we plotted the observations for which an increased $FR_{ANN} > 0.5$ was predicted by the ANNs on 2-D histograms in the triple-frequency space, colored by the median $FR_{ANN}$ of all the observations in the respective pixel. The pink line is drawn along the line of increasing median volume diameter expected for rimed particles according to Fig. 15 in Kneifel et al. (2015). With respect to frequency of occurrence (not shown here), the largest portion of the pixels for which a $FR_{ANN} > 0.5$

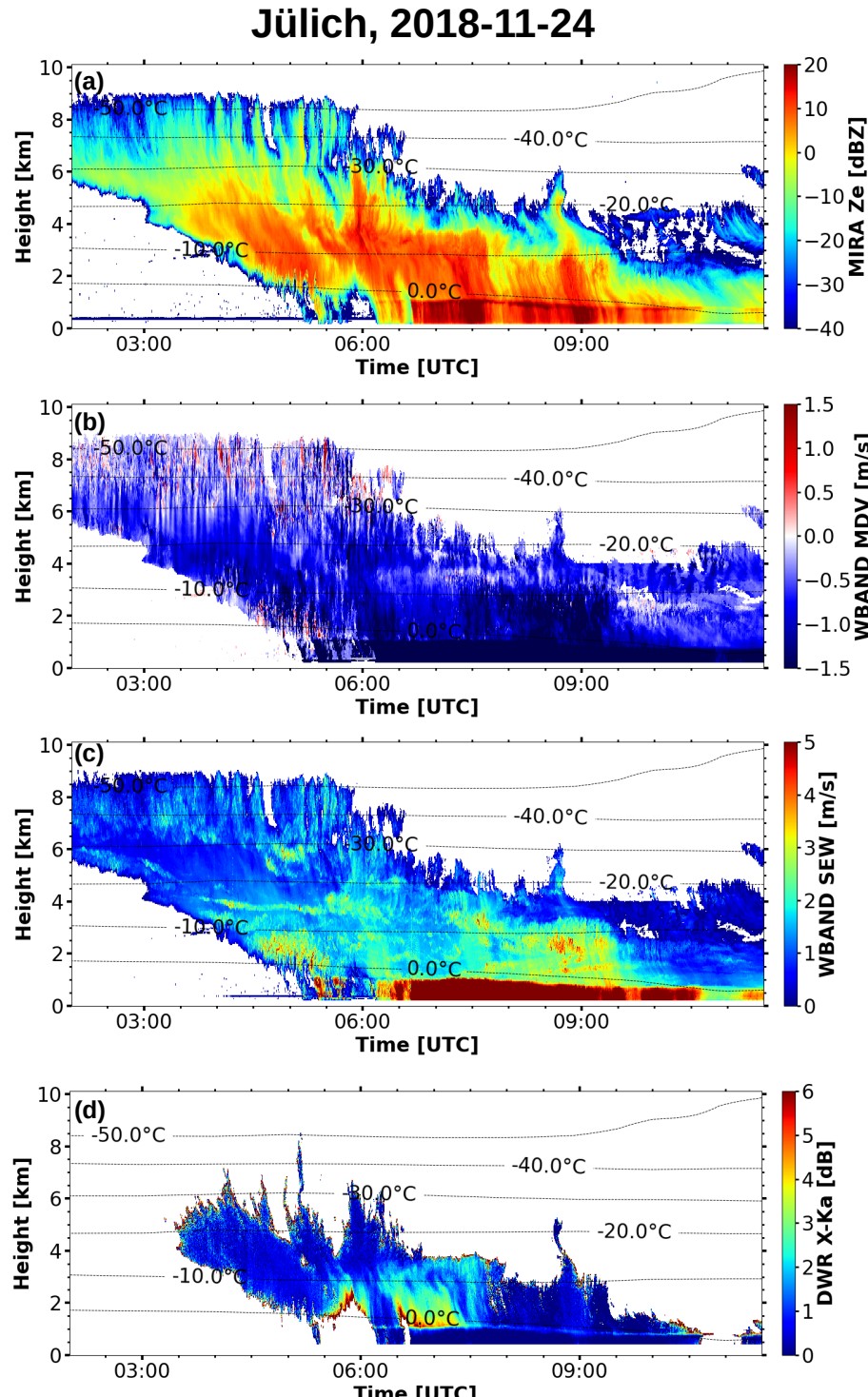

**Figure 5.** Radar moments measured during the 24 November 2018 case. a) Equivalent radar reflectivity measured by the Ka-band MIRA radar; b) mean Doppler velocity measured by the W-band radar; c) spectrum edge width derived from W-band Doppler spectra; d) Dual-Wavelengh-Ratio of X and Ka-band radar (DWR$_{X,Ka}$)

**Jülich, 2018-11-24**

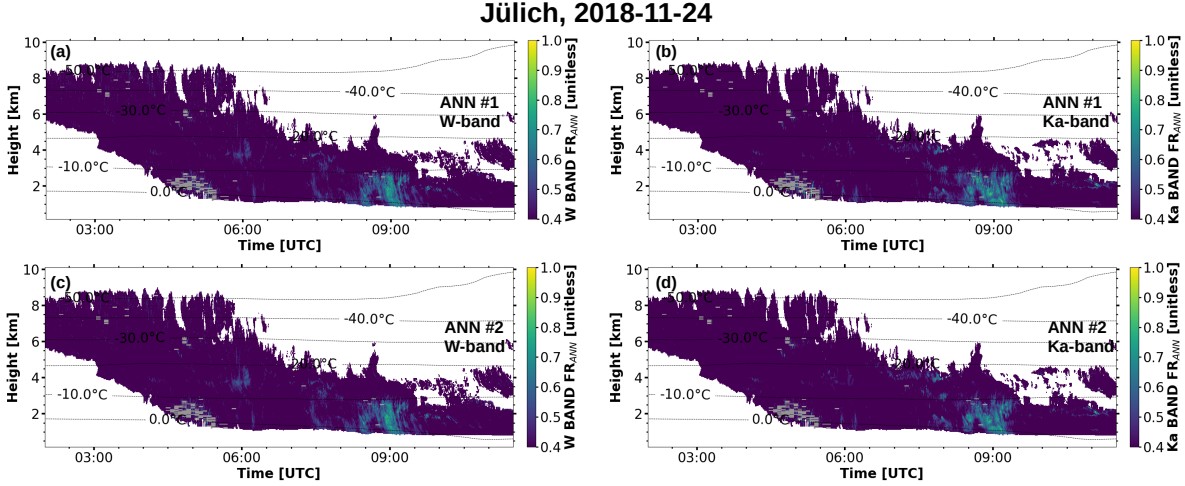

**Figure 6.** Predicted riming during the case on 24 November 2018 by ANN #1 and #2 using W-band moments (left column) and Ka-band moments (right column) as input features. The ANNs were applied only to those parts of the cloud which were classified as containing ice or ice and liquid by the CloudNet algorithm. Pixels with EDR above the threshold of $10^{-3} \mathrm{m}^2 \mathrm{s}^{-3}$ are masked in grey.

was predicted, falls around that line, for both ANN sets and both frequencies. This finding suggests that both ANN ensembles for both frequencies are capable of predicting elevated FR values.

### 3.3 Convective riming and aggregation case study in Leipzig

The findings in the previous sections motivate the need for additional comparisons to in situ observations. The 19 March 2021 Leipzig case is characterized by a wintertime convective mixed-phase cloud system, with cloud top at 3-4 km, and with

5 embedded strong snow and graupel showers. Cold air aloft combined with some solar warming near the ground causes weak instability. Additional lifting, e.g. due to convergences, triggers showers and, as in this case study, is even sufficient to cause thunderstorms.

In Fig. 8, the first two moments of the cloud radar Doppler spectra during that day are shown, along with the SEW. Around 15:00 UTC, a strong updraft is visible in the MDV, followed by strongly negative Doppler velocities coinciding with increased

10 Ze and SEW values. Around that time, alternating graupel and snow showers were observed in the Leipzig area. This case is moderately convective, and most of the data would be excluded by filtering criteria such as the convection index $\kappa$ used in the approach by Mosimann (1995).

Fig. 9 shows the ANN predictions for the time between 14:00 and 19:00 UTC. In the panels below, hydrometeors observed by the VISSS are shown for three selected time periods. In both ANN predictions, a clear transition from very high (around 1) to

15 very low ($\leq 0.4$) $\mathrm{FR}_{ANN}$ values at around 15:00 UTC is visible. This change can be confirmed by the VISSS observations: In the period from 14:40 to 14:50 UTC, the images show dense and roundish particles (rimed aggregates, graupel). In contrast,

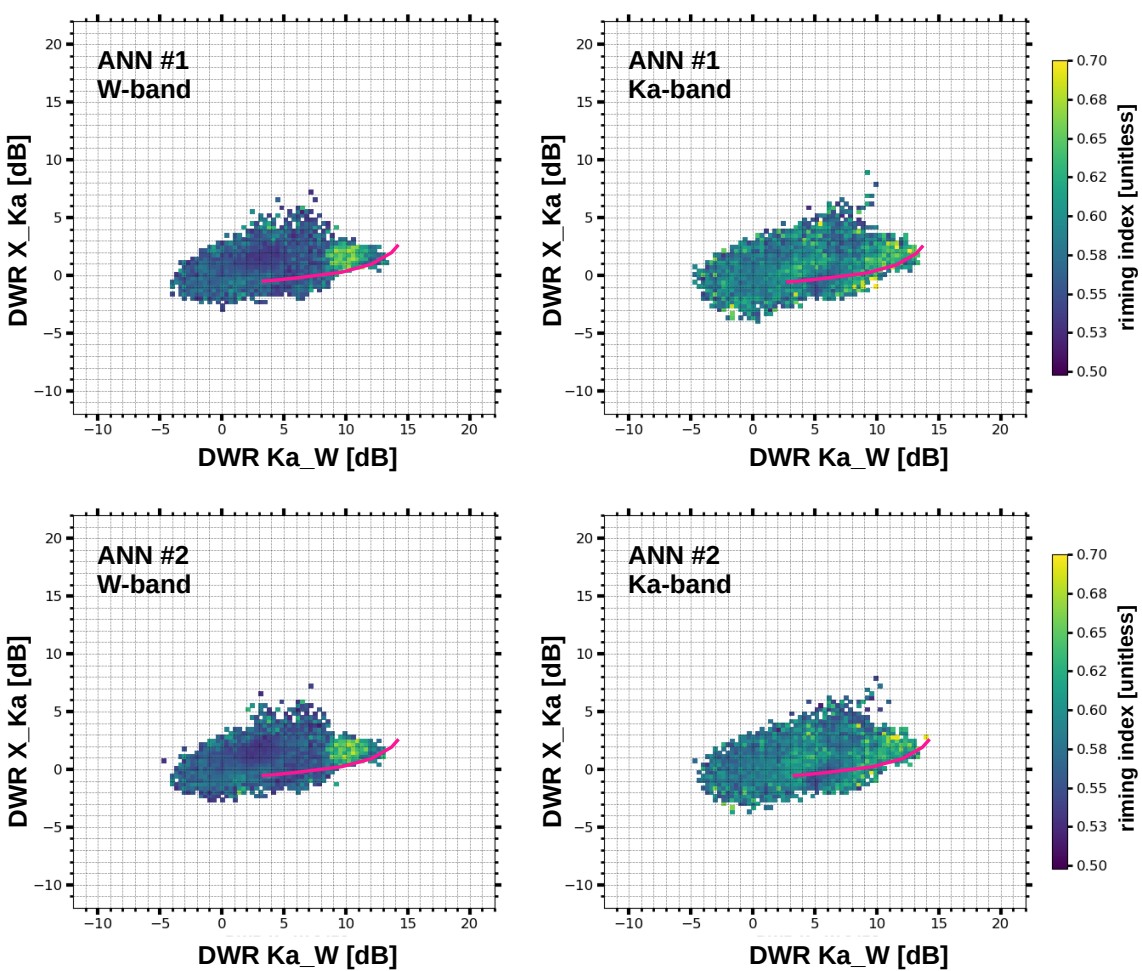

**Figure 7.** 2D histograms for seven cases chosen from the TRIPEx-pol data set where rime index $> 0.5$ was predicted. Each pixel contains at least 10 observations, and the color indicates the median FR$_{ANN}$ of all the observations contained in the pixel. FR$_{ANN}$ predicted using W-band and Ka-band radar observations are displayed in the top and bottom panel plots, respectively. The pink line is drawn where rimed particles are expected according to Fig. 15 in Kneifel et al. (2015).

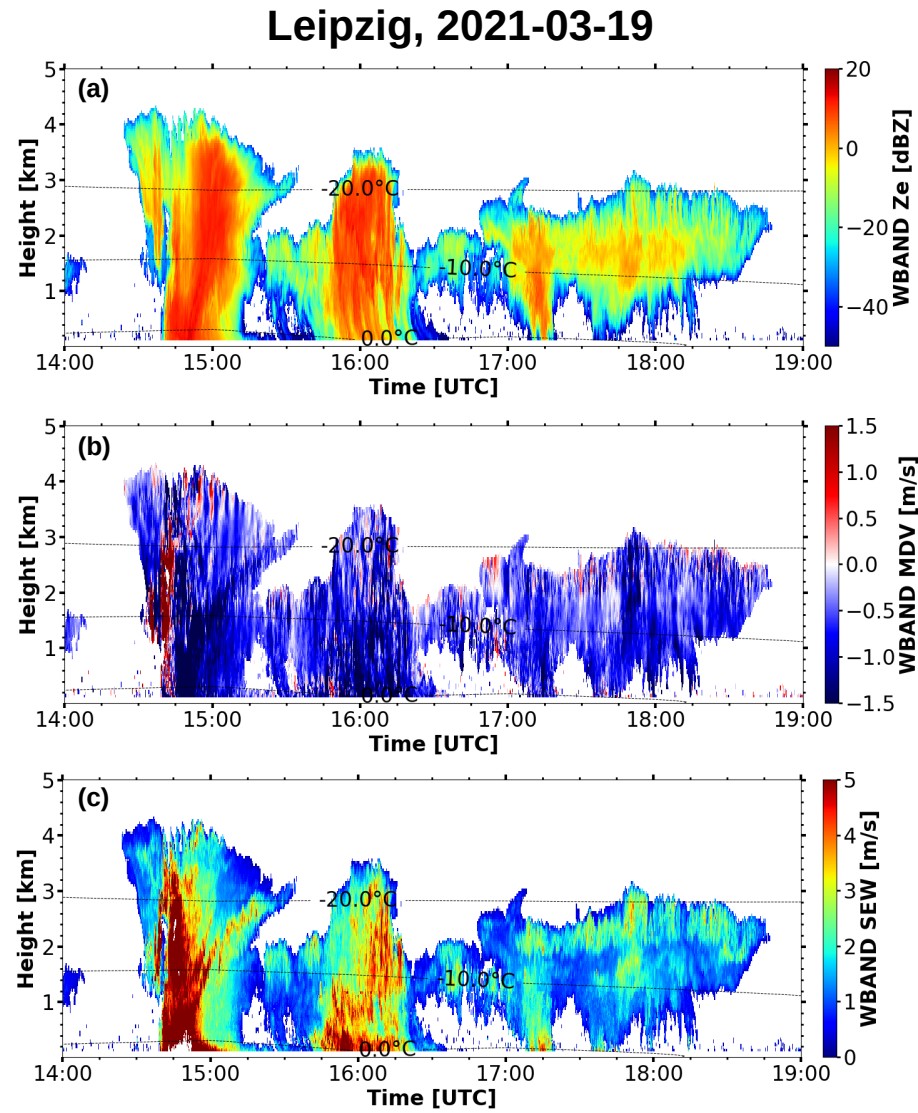

**Figure 8.** Radar moments measured by LIMRAD94 during the case on 19 March 2021. a) Equivalent radar reflectivity in dBZ; b) mean Doppler velocity; c) spectrum edge width

during the period from 15:00 to 15:10 UTC, fluffy, unrimed aggregates are observed. Later, from approximately 15:45 UTC to 16:20 UTC, the $FR_{ANN}$ features increased values throughout the vertical column ($\approx 0.7 - 1.0$), but not as pronounced as in the period before 15:00 UTC. VISSS images taken during the period from 15:50 to 16:00 UTC reveal a mixture of particles, which have different sizes and degrees of riming. Small hydrometeors with diameters well below 1 mm coincide with aggregates and graupel particles having sizes of several mm. The ANN predictions fit extremely well in line with these observations: In the first case, very high $FR_{ANN}$ values around 1 are predicted, whereas in the second case, the predicted riming indices are below 0.4. In the third case, even though a portion of the column is masked due to the EDR threshold, it is visible that intermediate $FR_{ANN}$ values are predicted by both ANNs. This leads to the assumption that the ANNs are not only capable of detecting strong riming, but are also sensitive to the degree of riming, or the fraction of rimed particles compared to the total hydrometeor population. In this case, as well as in the previously presented results, the predictions of ANN #1 and #2 are strikingly similar. These findings show that predicting riming is possible even without the use of MDV. It has to be acknowledged, though, that the vertical distribution of $FR_{ANN}$ cannot be verified by the VISSS observations. Since the ANNs proved to perform well for this wintertime convective case, this could open a door to detect and even quantify riming in convective systems.

### 3.4 Punta Arenas gravity wave case

The previous findings make us confident that ANN #2 can be applied to the W-band radar data in the DACAPO-PESO data set. We do not apply ANN #1, because it would be biased by the orographic wave motions. Here, we analyze a case observed on 21 February 2019 (Fig. 10) from 13:30 to 22:00 UTC. A precipitating cloud system with cloud top around 2.5 to 3 km is present from around 15 to 18 UTC. Above, a mid-level cloud with top around 6 km and varying cloud base is observed. Especially in the higher-level cloud, in the range between 3 and 6 km, a wave pattern is visible in the MDV (Fig. 10b), including a prominent downdraft around 18:00 UTC with MDV $\approx -2$ ms$^{-1}$. Precipitation was reaching the ground between 15:00 and 18:00 UTC, and another precipitation event occurred around 21:00 UTC. At 16:30 UTC, marked by the red cross in Fig. 10d, graupel particles were observed at the ground on-site.

In Fig. 10d, the $FR_{ANN}$ predicted by ANN #2 is shown. The predicted $FR_{ANN}$ is only increased in the lower part of the cloud system up to around 3.5 km range, which probably contains liquid water. No increased $FR_{ANN}$ values are predicted for the higher part of the cloud system. The maximum in $FR_{ANN}$ predictions is reached around 16:30 UTC, coinciding with the observation of graupel particles.

### 4 Summary, Conclusions and Outlook

In this work, we have demonstrated the ability of artificial neural networks (ANNs) to estimate riming using ground-based zenith-pointing cloud radar measurements as input features. Training data were extracted from the BAECC data set by temporally matching PIP-based riming retrievals with cloud radar observations at cloud base. Ensembles of ANNs were trained to predict a $FR_{ANN}$, separately for Ka-band and W-band observations, using three different combinations of input variables: ANN #1 uses the equivalent radar reflectivity factor (Ze), the mean Doppler velocity (MDV), the spectrum edge width (SEW)

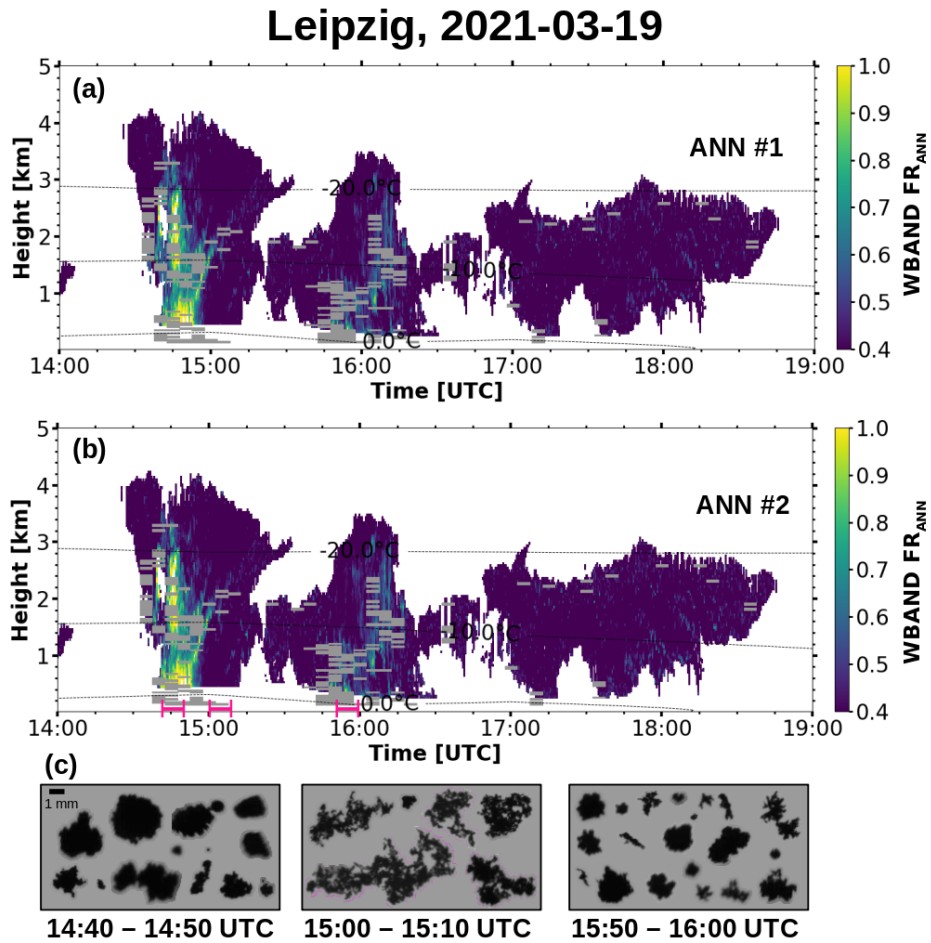

**Figure 9.** Riming during the case on 19 March 2021 predicted by (a) ANN #1 (SEW, Ze, skewness, MDV), (b) ANN #2 (SEW, Ze, skewness). The panels in (c) show images taken by the VISSS during the period from 14:40 to 14:50 UTC, the period from 15:00 to 15:10 UTC, and the period from 15:50 to 16:00 UTC. These periods are marked on the time axis in (b) with pink lines. Pixels with EDR above the threshold of $10^{-3} m^2 s^{-3}$ are masked in grey in (a) and (b).

# Punta Arenas, 2019-02-21

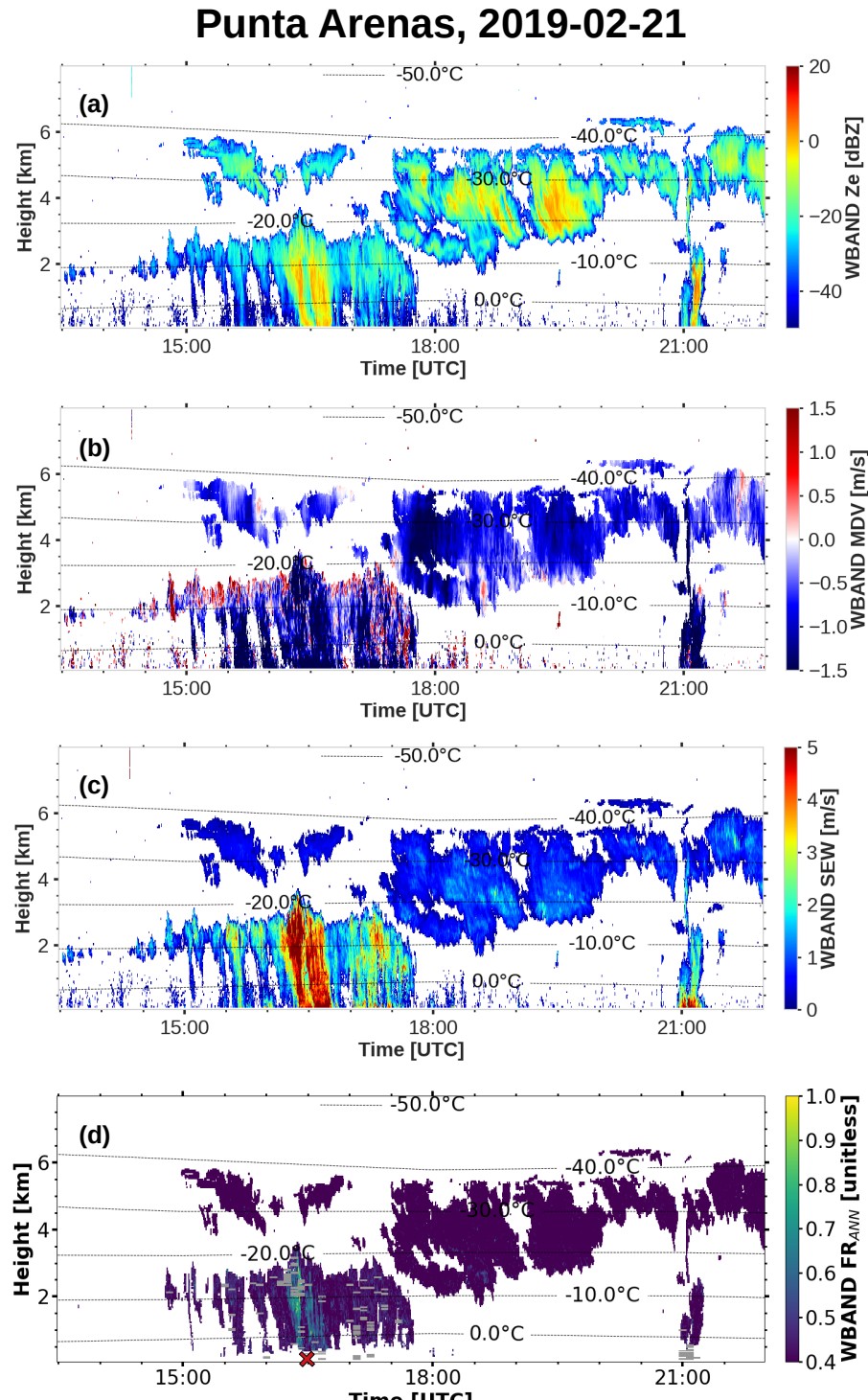

**Figure 10.** Radar moments measured by LIMRAD94 during the case on 21 February 2019. a) Equivalent radar reflectivity in dBZ; b) mean Doppler velocity; c) spectrum edge width and d) FR$_{ANN}$ predicted by ANN #2 (using, Ze, SEW and skewness). The red cross marks the time (16:30 UTC) when graupel particles were observed at the site.

and skewness as input features; ANN #2 uses Ze, SEW and skewness. One set of ANNs using only Ze and MDV as input features was not further considered due to low performance indicating that these two quantities are not sufficient for quantifying riming. We evaluated the trained models using four case studies and a longer data set comprising observations of seven mixed-phase cloud systems. In general, the predictions of ANN #1 and #2 were found to be very similar across all considered cases despite the different input variables. Both ANNs were able to predict strong riming and capture the subsequent transition to unrimed snow reported in literature for a case from the BAECC experiment (Fig. 4). A turbulence threshold of EDR = $10^{-3}$ $m^2s^{-3}$ was applied to prevent too high $FR_{ANN}$ ANN estimates due to broadened Doppler spectra. It was shown that the models are able to generalize to a new data set, i.e. different radar systems for both considered wavelengths (Fig. 6), and different meteorological conditions. ANN predictions for seven cloud cases were shown to match expected signatures of riming in the triple-frequency observation space of X-, Ka- and W-band (Fig. 7). Large $FR_{ANN}$ values mostly fall into the region for which riming is expected. The application of both ANNs to a convective wintertime cloud case showed that the method can also be applied to convective systems (Fig. 9). Because ANN #2 does not depend on MDV, it was applied to an orographic case, yielding high $FR_{ANN}$ values for the period during which solid graupel particles were observed at the site (Fig. 10). These findings indicate that retrieving riming is possible even without the use of MDV.

One of the major constraints of this study is the limited training data set. As better and longer-term training data sets become available, the ML techniques at hand can be more fully exploited and further improvement of the ANN performance is expected. Also, an extended data set would better allow to quantify the errors of the ML method and to understand the limitations with respect to identifying very high and very low FR values.

This study closes an important gap in our abilities to quantify the riming process with cloud radars. Further validation, e.g. by comparison of this technique with airborne in situ observations would be a useful extension of this work. Future applications will focus on longer-term data sets to investigate the drivers of riming, including orographic conditions.

*Code and data availability.* The ARM data used in this study are freely available on the ARM data discovery portal:

Atmospheric Radiation Measurement (ARM) user facility. 2014. Ka ARM Zenith Radar (KAZRSPECCMASKMDCOPOL). 2014-02-21 to 2014-02-22, ARM Mobile Facility (TMP) U. of Helsinki Research Station (SMEAR II), Hyytiala, Finland; AMF2 (M1). Compiled by I. Lindenmaier, N. Bharadwaj, K. Johnson, D. Nelson, A. Matthews, T. Wendler and V. Castro. ARM Data Center. Data set accessed 2021-05-07 at http://dx.doi.org/10.5439/1095603.

Atmospheric Radiation Measurement (ARM) user facility. 2014. Microwave Radiometer (MWRLOS). 2014-02-21 to 2014-02-22, ARM Mobile Facility (TMP) U. of Helsinki Research Station (SMEAR II), Hyytiala, Finland; AMF2 (M1). Compiled by M. Cadeddu. ARM Data Center. Data set accessed 2021-05-11 at http://dx.doi.org/10.5439/1046211.

Atmospheric Radiation Measurement (ARM) user facility. 2014. Balloon-Borne Sounding System (SONDEWNPN). 2014-02-01 to 2014-03-20, ARM Mobile Facility (TMP) U. of Helsinki Research Station (SMEAR II), Hyytiala, Finland; AMF2 (M1). Compiled by E. Keeler, R. Coulter and J. Kyrouac. ARM Data Center. Data set accessed 2021-08-10 at http://dx.doi.org/10.5439/1021460.

CloudNet data used in this article are generated by the European Research Infrastructure for the observation of Aerosol, Clouds and Trace Gases (ACTRIS) and are available from the ACTRIS Data Centre using the following link: https://hdl.handle.net/21.12132/2.83f2cbd65d504b36.

The PIP data are available at https://github.com/dmoisseev/Snow-Retrievals-2014-2015 and the training data set at https://github.com/ti-vo/BAECC_features

Code used for the data analysis is freely available on GitHub: https://github.com/ti-vo/riming_detection_ML/tree/master

*Video supplement.* We are uploading a 3D animation plot to the TIB AV Portal

*Competing interests.* We declare that we have competing interests as follows: Maximilian Maahn and Stefan Kneifel are associate editors of AMT.

*Acknowledgements.* We acknowledge the provision of physical access to the LACROS resources in the frame of DACAPO-PESO, which is provided via the European Research Infrastructure for the observation of Aerosol, Clouds and Trace Gases ACTRIS under grant agreement no. 654109 and 739530 from the European Union's Horizon 2020 research and innovation programme. We also want to thank Boris Barja from the Universidad de Magallanes for granting access to the site and his support throughout the field campaign.

We acknowledge ACTRIS for providing the CLU (2021) dataset in this study, which was produced by the Finnish Meteorological Institute, and is available for download from https://cloudnet.fmi.fi/. DWD for providing ICON model data.

We gratefully acknowledge the funding of the German Research Foundation (DFG) in the frame of the special priority program on the Fusion of Radar Polarimetry and Atmospheric Modelling (SPP-2115, PROM, Grant KA 4162/2-1)

Work provided by S. Kneifel was funded by the German Research Foundation (DFG) under grant KN 1112/2-1 as part of the Emmy-Noether Group OPTIMIce. The TRIPEx-pol campaign has been supported by the DFG Priority Program SPP2115 "Fusion of Radar Polarimetry and Numerical Atmospheric Modelling Towards an Improved Understanding of Cloud and Precipitation Processes" (PROM) under grant PROM-IMPRINT (Project Number 408011764).

T. Vogl acknowledges funding from the German Academic Exchange Service for a research stay at CU Boulder, Colorado (Grant 57504619)

We would like to thank Leonie von Terzi for her help processing the TRIPEx-pol data set.

A big thank you to Martin Radenz, who saw the graupel in the grass in Punta Arenas after coming back from lunch break, and wrote it into the logbook.

We also want to thank David John Gagne from NCAR for his advice on machine learning.

Thank you to Anton Kötsche for the synoptic analysis, and special thanks to Jen Kay and Patric Seifert for fruitful discussions.

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
