# Peer review of "Using artificial neural networks to predict riming from Doppler cloud radar observations"

_Atmospheric Measurement Techniques, 2021_

## Author Response (AR1)

We want to thank the editor and the two anonymous referees for their careful reading of our work and their comments, which certainly helped to improve our manuscript. Furthermore, the resulting discussion has given us inspiration to improve our study even beyond the required points.

We are addressing the raised comments in a point-by-point way below. We have organized the reviewer comments in a manner such that RxSn represents the nth specific comment of referee x, and RxTn the nth technical comment by reviewer x. We hope this will provide a clear basis for discussion during the further reviewing process.

We are providing a brief overview of the most important changes we have made to our study:

1. Inclusion of W-band data from the BAECC data set as a separate training data set, to ensure that the trained models can be applied to the Punta Arenas and Leipzig data sets where only W-band radar observations are available.

2. Attenuation correction for all considered data sets (gases, liquid, melting layer, ice)

3. Application of a turbulence threshold (EDR = $10^{-3}$ $m^2 s^{-3}$), which was set after a sensitivity study to make sure that spectral broadening due to turbulence does lead to false predictions of riming

4. Changes to the sampling technique of the training data, applying this turbulence threshold instead of sampling at CBH. This has the advantage that we can sample at lower altitudes, closer to the radar, ensuring better spatio-temporal matching of in-situ and remote sensing observations. As a consequence, all ANNs had to be retrained.

5. Treatment of the scaler, which is used to scale the training data features to a value range between 0 and 1, as part of the model. This means that the scalers defined for Ka-band and W-band respectively on BAECC data were also applied to TRIPEx-pol, DACAPO-PESO and Leipzig data. Previously, each instrument had its "own" scaler. This change led to a decrease in predicted FR values, but the features remained similar.

6. Bug fix: We accidentally used kurtosis instead of skewness in the TRIPEx-pol data set. This changed the result plots in Section 3.2, e.g. removed the "hook feature" in the triple-frequency plots.

**Comments by the editor**
Comments to the Author:
Dear authors,

your manuscript is ready for publication in AMTD.
I, however, have some minor technical comments to the manuscript, which I kindly ask you to consider (in addition to the referee comments) in the review phase of the manuscript.

P1, L1: Introduce SLW
P4, L10: Introduce SEW (only appears in the abstract, so far)
P10, L13: 17.00 --> 17:00
P15, Fig. 5: Please correct the text in Figures b and d. Also for the right column 'W-band' is indicated. But it should be Ka-band, shouldn't it?

We want to thank the editor for the comments. We have made the suggested changes to the manuscript.

**Comments by Reviewer #1**

This work describes the development and evaluates the performance of artificial neural networks (ANNs) to diagnose riming using measurements from vertically-profiling Doppler radars.

This is a useful scientific objective. From a remote sensing perspective, knowing when and where riming is occurring can help improve estimates of precipitation rates and reduce uncertainties in retrievals based on radar observations. The work suggests that riming can be diagnosed without using mean Doppler velocity - this is useful because mean Doppler velocity will be perturbed in areas of ascent or descent such as convective cores or atmospheric waves. Finally, detecting and quantifying riming provides information about the microphysical processing experienced by ice-phase precipitation.

The method for development of the ANNs is sound, as is the testing of the ANNs, but I feel there are some shortcomings in terms of scientific quality. The analysis of the test results, particularly for the BAECC cases, seems unnecessarily limited and basic. While several sources of uncertainty are acknowledged, there is only a rudimentary discussion of quantified uncertainties, and there is no quantitative estimate of the propogation of these uncertainties into the results of the study. Additionally, one critical source of uncertainty is not addressed.

Figures are generally well-made and used well to convey information. The narrative overall is good, but there is some text that would benefit from better organization. See my specific comments below for details. I think it would be worthwhile to provide more quantitative analysis of the BAECC results and make some minor reorganization to the section that is noted below. I'm recommending this be treated as a "major revision".

Specific comments
##################
**R1S1)** The method for creating the FR values in the BAECC dataset depends on a mass-dimension relationship for unrimed snow, and neither the uncertainty of this relationship nor its impact on the FR predictions of the ANN are assessed. The mass-dimension relationships for snow particles are highly variable.

We want to thank the reviewer for pointing us to this possible source of error. It is true that there is a large variability in m-D relations reported in literature. However, the variability in this relation for unrimed snow is considerable smaller than for rimed particles. Moisseev et al. (2017) compare a selection of published m-D relationships (Fig. 2 in their publication), and conclude that their relationship to derive the mass of unrimed snow "overestimates unrimed snowflake masses by a maximum of 5%". We have taken this into account and are addressing this point more thoroughly together with **R1S9** (see below).

**R1S2)** The description of the development of the ANN provided in section 2.3.2 would benefit from some reorganization. See my comments below about some explanations about the hyperparameters and loss function that seem to be out of order. In the discussion of the testing with the TRIPEx-pol data, there's very little in the way of quantitative comparisons between the results for ANN #1 vs. #2 and W-band vs. Ka-band. The description of the data (section 2.1.2) indicated that the data were regridded onto a common height-time grid. Wouldn't it be possible to make more quantitative comparisons, for example by looking at whether there is a 1:1 relationship among the four possible combinations? I

think a quantitative comparison would be a more convincing demonstration rather than the statements that the "predicted riming index is very similar and the features are almost identical" (P13, L13-14).

Thank you for this useful suggestion. The proposed comparison between Ka- and W-band for ANN #1 and #2 is indeed relatively easy to accomplish for the TRIPEx-pol data set.
Below, Fig. 1 shows scatterplots of the predicted $FR_{ANN}$ of W-band vs. Ka-band for ANN #1 (left) and ANN #2 (right). Your presumption that there should be a 1:1 relationship (dashed line) holds for both ANN sets, as a large portion of the dots falls close to this line. Please note the logarithmic color scale. The $R^2$ values are 0.81 for ANN #1 and 0.73 for ANN #2. Instead of adding this figure to our study, we propose to add the following to section 3.2 :

*"Furthermore, the common time-height grid in the TRIPEx-pol data set allows for convenient direct comparison of the two radar frequencies for each of the ANN set ups:  The correlation of the predicted $FR_{ANN}$ values is high for both ANN #1 and #2, the $R^2$ being 0.73 and 0.81, respectively."*

[Figure]

*Figure 1: scatterplots of Ka-band and W-band ANN predictions for the TRIPEx-pol data set, for ANN #1 (left) and ANN #2 (right)*

**R1S3)**  P3. L2-3: It would be helpful to be clear how "terminal fall velocity" is being defined. Is this the fallspeed (speed at which a particle approaches the Earth's surface) or is the terminal speed (or terminal "velocity", which is relative to the motion of the fluid)? Also, in almost no case will the speed of a single particle be equal to MDV, since MDV is determined by an integration over the Doppler spectrum.

Thank you for spotting this inaccuracy. You are right, we did not mean the terminal speed but the fall speed. We are proposing to rephrase the sentence as follows:

*"The method fails when the assumption that the MDV is equivalent to the particle fall speed in the observation volume does not hold."*

**R1S4)** P4, L5: It is not fully clear to me what is meant by "developed models". Is this referring to the ensemble of ANNs?

We acknowledge that this choice of words may be confusing to the reader. We propose to change this phrase to *"the ensembles of trained ANNs"*

**R1S5)** P5, L9-10: Can you describe why these particular days were selected?

These are cases of thick mixed-phase cloud cases, both precipitating and non-precipitating, observed during TRIPEx-pol. To limit computation time while still covering a large range of meteorological conditions encountered during the campaign, a selection had to be made in consultation with the responsible scientists (Stefan Kneifel, Leonie von Terzi, José Dias Neto). We propose to add *"all featuring mixed-phase cloud cases, possibly with riming."*

**R1S6)** P6, L8: It is not clear here what is meant by "having the same D_max as the observed particles". Since IWC is determined by integration over the size distribution, it is not necessary that the sizes of the observed and unrimed snow particle be the same. Do you mean "for unrimed snow having the same N(D_max) as the observed particles"?

Yes, you are correct. We included the proposed change (N(D_max) instead of D_max) in the revised version of our manuscript.

**R1S7)** P6, L9-10: What was the total number of samples examined, and how many of those had FR < 1?
Thank you for this comment. Is it possible you mean FR < 0 (and not 1)?

FR < 0 values can result if the retrieved masses are below the masses assumed for unrimed snow.

number of PIP data points: 789

number of FR < 0: 138

We acknowledge that our training data set is limited with regards to the number of valid data points. However, the BAECC data set is, to our best knowledge, the longest data set available including both high-quality cloud radar data and in situ observations of snow. This study is aimed at **demonstrating** the **ability** to train ANNs to predict riming using cloud radar observations as input features. Significant improvements of the performance is expected when better/ longer training data are available.

**R1S7)** P6, L15-16: Since air below CBH is subsaturated, is the loss of ice mass by sublimation possibly significant?

Thank you for this comment. We considered the surface RH measurements coinciding with our sampled training data. The vast majority of the data points has RH of 85% and more, thus we assume sublimation to be negligible.

**R1S8)** P6, L18: Was it necessary to assume that MDV is constant between CBH and the surface? Doesn't the KAZR provide MDV profiles at bins below CBH?

We agree that making use of the MDV observations below the range at which radar observations were sampled would make the sampling method more accurate, however, also a lot more complex. We decided to rely on this more simple approach due to the comparably coarse PIP time resolution of 5 minutes.

If we consider the range at which radar data are sampled, on average at 247 m and at the maximum 940 m range, and assume 1 m/s MDV, 940 m sampling height translates to 940 s time offset correction. If more accurate tracking of the MDV below sampling height would result in a velocity variation of 10%, the time for the particles to reach the surface would vary by ~ 105 s, which is significantly smaller than the PIP time resolution of 5 mins. We thus assume that this less accurate time offset correction does not strongly impact the matching of radar and PIP measurements.

**R1S9)** P7, L10-15: Then, how are the impacts of these uncertainties quantified? Is this discussed later in the paper?

We acknowledge that our manuscript is missing a thorough error consideration. In addition to the error in the mass of unrimed snow discussed in the answer to **R1S1**, we performed a sensitivity study to get a grasp of the effect of measurement uncertainties on the ANN predictions. We propose to add a section discussing these sources of uncertainties as follows:

*"2.4 Error consideration*
*We acknowledge that several sources of uncertainty impact the riming predictions, only some of which can be quantified. For the training data set, assumptions about the density of unrimed snow and the viewing geometry corrected $D_{max}$ were made. The m-D relation used to derive the mass of unrimed snow (Moisseev et al., 2017) is assumed to overestimate $m_{us}$ by a maximum of 5%. Furthermore, errors are introduced by the spatio-temporal matching of radar and PIP observations.*
*When the trained ANNs are used to predict riming from other radar observations, measurement errors propagate into the $FR_{ANN}$. We performed a sensitivity study to get a grasp of the overall effect of measurement uncertainties on the ANN predictions. We assumed an error of 0.2 dBZ for Ze, one Doppler bin for MDV and SEW, and 0.2 for the skewness, and added this uncertainty to selected Doppler spectra. This was accomplished by random picking of n=10,000 samples from a Gaussian distribution defined by the measured values as mean and above errors as standard deviation. The resulting standard deviation in the predicted $FR_{ANN}$ is approximately 1%. In addition to the 5% uncertainty from the mass of unrimed snow, and other error sources which cannot be quantified we propose to assume an overall uncertainty of approximately 10% due to the training data. This uncertainty has to be added to the uncertainty of the ANN models (Table2)."*

**R1S10)** P7, L24: Not just moments, but also SEW, which is not a moment.

You are right, we added the proposed change.

**R1S11)** P8, L7: Is the use of "x" in "max(0,x)" intentional, and does it represent time as it did in P6, L16?

Thank you for spotting this. We acknowledge that the double use of "x" may be confusing to the reader and propose to change the first "x", i.e. the time correction, to "Δt" in both the text and Figure 1.

**R1S12)** P8, L9-10: Could you provide a more complete description of the error? I'm not sure of the meaning of "This error is defined by the loss function". Is this simply a measure of the difference between the outputs of the ANN and the actual observed values? How is it calculated?

Yes, the error is simply the squared error, which is the most common loss function used for regression problems. We propose to add
 *"in our case the squared error"*
to improve the comprehensibility of our text.

**R1S13)** P9, L1-7: This paragraph begins by identifying the number of hidden layers and number of neurons as hyperparameters to be tuned, but the following description of k-fold cv says nothing about how these hyperparameters are tuned. Instead, the role of k-fold cv to produce multiple models for estimating average and variance is described.

We are addressing this point together with **R1S15** below.

**R1S14)** P9, L16: Does "layer depth" = number of hidden layers?
Yes. We propose to add *"(which is also referred to as the layer depth)"* at the point where we mention the number of hidden layers in the beginning of this paragraph.

**R1S15)** P9, L16-19: This seems to be the content that should follow the first sentence of the paragraph starting at P9, L1 (see my comment above).

We agree, this part should be shifted upwards. Thanks for the suggestion.

**R1S16)** P9, L17: The RMSE of what?
Of ANN predictions and target values in the training/ validation set. We are adding this to the text.

**R1S17)** P9, L20: So, is this the definition of the "loss function" (see my earlier question) and is the RMSE described above on L17?

No, this refers to the testing set. The loss function, i.e. the squared error, is used internally in the (Python sklearn) model framework. The RMSE of the training/ validation set is used for tuning the hyperparameters. This paragraph now refers to the testing phase, where we manually compared the RMSEs of the ANN predictions and the testing set targets. We decided to compute the RMSE separately for all FR values and high (>0.5) FR values, to choose which models could be applied to other data sets in the next steps.

**R1S18)** P9, L22-24, P10, L1-2: There are a few point to ask regarding this paragraph. What is meant specifically by "less well represented"? How can high FR cases be "less well represented" but still be "clearly separated in the input feature space"? The RMSE *is* the error, or at least it is one way of measuring the error. It is probably not necessary to say that it "basically represents the error". Is the performance of ANN #0 actually "much worse"? The "test set RMSE" is only 0.02 larger than that for ANN #2.

We want to thank the reviewer for this remark. Regarding the first point: What we were trying to say here is that our input data set is not uniformly distributed with respect to FR. Cases with high $FR_{PIP}$ (>0.6) are rare compared to cases with $FR_{PIP}$ between 0.2 and 0.6. However, those cases with high $FR_{PIP}$ are separated in the multidimensional input feature space, as can e.g. be seen in the 3D figure inset in Fig. 1 in our manuscript. We agree that the text could be improved in terms of clarity, and should be updated to include the SMOTER technique, which we applied in the revision to mitigate this problem. We propose to add the following to Section 2.3.1:

*"This skewed distribution of target values has impacts on the training of machine learning algorithms and needs to be taken into account carefully. ANN models will primarily be trained to predict values in the range where most of the observations lie. In order to ensure that ANNs are trained to predict high values of FR, which are of interest, the training data set is resampled using the synthetic minority oversampling technique for regression problems (SMOTER, Torgo et al., 2018). SMOTER is an algorithm which oversamples rare cases and undersamples frequent cases in the training data set, leading to a more balanced distribution."*

Regarding the second point: The table has been updated, due to different training data sampling **(4)**, the inclusion of the MWACR radar **(1)**, and application of SMOTER. However, now, the RMSE for ANN #0 is even smaller than for ANN #2. The RMSE does not seem to give a strong indication that ANN #0 performs worse than the other two ensembles. We made the decision to exclude ANN #0 after inspecting the RMSE, as well as time-height plots of predicted $FR_{ANN}$ and scatter plots of ANN performance on the test set. Fig. 2 shows scatter plots (colored by density) for the different ANN ensembles, for both Ka and W-band, on the testing data set. We are also including a time-height plot of $FR_{ANN}$ for the BAECC riming case in Fig. 3. It is visible in both the scatter plot and the time-height plot that ANN #0 (trained on MWACR data) seems to only distinguish between two 'modes', low values around 0.4 and higher values around 0.65. The other two set ups are able to yield a range of FR values.

It is obvious that ANN #0 performs "much worse" than the other two ensembles, however, this observation is not reflected by the RMSE. In another attempt to put numbers on this observation, we fit linear models to the scatter plots of the testing data set. The slope should ideally be close to 1. The slopes (red dashed lines in Fig. 2) are listed in Table 1. At both considered frequencies, the results for ANN #0 are lower than for the other two ensembles.

We propose to include Fig. 2 and discuss and justify our decision to let go of ANN #0 in the paper as follows:

*"In the testing phase, the model performance is evaluated using the test set prediction RMSE, in connection with visual inspection of scatter plots of ANN test set predictions and target values. In addition, time-height plots of ANN predictions are examined with regard to physical plausibility of the predicted features. We decided to put an additional focus on the ability of the networks to predict high FR values > 0.5. The reason for this choice is that the extracted Doppler spectra features are expected to contain little information about riming up to a moderate riming stage. For high FR values, the prediction should be more accurate because the riming signal should be clearer in the input features. Table 2 summarizes the test set RMSEs found for the three different ANN input parameter sets and the two radar frequencies. Fig. 2 shows scatter plots of ANN predictions and test set target values, for the three different ANN ensembles, for Ka- and W-band. The slopes of the linear models (red dashed lines in Fig. 2) are listed in Table 2. From Fig. 2 it becomes apparent that ANN #0 seems to have a problem in the W-band set up, because two populations of values are separated in the predictions. The slopes of*

*the linear fits, which should ideally be 1, are lowest for ANN #0 for both considered radar frequencies. This observation is however not reflected in the RMSEs listed in Table 2, which hardly differ between the three ANN ensembles for the W-band. The RMSEs in turn show that ANN #0 performs worse than the other two setups for the Ka-band. This illustrates the limits of using a single quantity like RMSE as a quality metric. ANN #1 and #2 also have issues predicting high FR values, and all ANNs feature a cut-off at low FR values. This might be because riming signatures for low FR are not very pronounced in the cloud radar Doppler spectra, making it impossible for the ANNs to accurately predict FR < 0.3 from the extracted features. For the remaining study, we will focus only on ANN #1 and ANN #2"*

Finally, we propose to remove the sentence *"The RMSE basically represents the error… "* because as the referee correctly noted, it does not make any sense.

[Figure]

*Figure 2: scatterplot of ANN performances on the test set. The black dashed line marks the 1:1 line, the red dashed line is a linear model fit to the data. (a) W-band, ANN #0; (b) W-band, ANN #0, only high FR$_{PIP}$ values; (c) Ka-band, ANN #0; (d) Ka-band, ANN #0, only high FR$_{PIP}$ values; (e) W-band, ANN #1; (f) W-band, ANN #1, only high FR$_{PIP}$ values; (g) Ka-band, ANN #1; (h) Ka-band, ANN #1, only high FR$_{PIP}$ values; (i) W-band, ANN #2; (j) W-band, ANN #2, only high FR$_{PIP}$ values; (k) Ka-band, ANN #2; (l) Ka-band, ANN #2, only high FR$_{PIP}$ values*

|  | W-band | Ka-band |
|---|---|---|
| ANN #0 | 0.41 | 0.23 |
| ANN #1 | 0.46 | 0.38 |
| ANN #2 | 0.46 | 0.32 |

[Figure]

*Figure 3: time-height plot of predicted FR$_{ANN}$ by ANN #0, trained on MWACR data.*

**R1S19)** P10, L2: Please clarify this. Why would you change the name for a quantity because there are many steps involved in its calculation?

We want to thank the reviewer for this comment. The second referee had similar concerns, and we acknowledge that renaming the quantity may cause a lack of clarity. After further consideration, we want to propose to rename the "riming index" to FR$_{ANN}$, i.e. the FR predicted by the ANNs, as opposed to FR$_{PIP}$, the FR measured by the PIP. This clearly separates the predicted from the measured FR, while making clear that the quantity is the same.

**R1S20)** P10, L12-13: I expect this 1.5 hour period on 21 Feb 2014 was not included in the training data. Is that correct? Is it part of the 10% of data that was retained for the testing phase (see P7, L28-29.

Yes, this is correct. We propose to add *"which is part of the 10 % of the labeled data that were retained for the test set."* to make this point clear to the reader. Please also note that the 1.5-hour period was changed to a 1-hour period (22-23 UTC) due to strong attenuation of the MWACR data.

**R1S21)** P13, L20-21: The differences in sensitivity between the W- and Ka-band radars have not been discussed previously (section 2.1.2). I'm not sure that I understand this comment about the differences in sensitivity. For either the W-band or Ka-band radar, if the backscattering from a radar bin is below the minimum detectable signal of the radar, I would guess that there is no data for that radar bin and there would be no ANN estimate of FR for that radar bin. How would *sensitivity* influence the *accuracy" of the predicted FR?

We want to thank the reviewer for this comment. We have apparently mixed up the radar "sensitivity", i.e. the minimum detectable signal, with the noise level. We were meaning to say that the predicted FR is similar for the Ka and the W-band radars, despite the systems having different scattering and noise properties and being impacted differently by attenuation due to their different wavelengths. We are proposing to rephrase the paragraph as follows:

*"$FR_{ANN}$ is similar for the Ka and the W-band radars, despite the systems having different scattering and noise properties and being impacted differently by attenuation due to their different wavelengths. "*

**R1S22)** P13, L24-25: I think this needs some additional details. You seem to be describing the features of a plot, but it is not clear what is being plotted. Also, regarding "These signatures are attributed to changes in the particle density during the riming process", I assume that only applies to the riming signature. Is there an explanation also for the "hook-like" signature associated with aggregates?

Thanks for this comment. We agree that this part of the manuscript could be improved with respect to comprehensibility. We propose to change the section as follows:
*"As mentioned earlier on, riming and aggregation can produce distinct signatures in the triple-frequency space of X-, Ka- and W-band reflectivity. When considering a plot with $DWR_{X,Ka}$ on the abscissa and $DWR_{Ka,W}$ on the ordinate, observations obtained during riming events fall onto a line at low $DWR_{X,Ka}$ (pink line in Fig. 7). Aggregates, in contrast, tend to yield a hook-like feature, due to their comparably large size at which $DWR_{Ka,W}$ is in saturation because of Mie scattering. This conceptual model was presented by Kneifel et al. (2015) and further explored by Mason et al. (2019) who found that not only the density, but also the shape parameter of the particle size distribution and the internal structure of aggregates can have strong impacts on triple-frequency signatures."*

**R1S23)** P15, L1-2: I see a (weak) suggestion of a hook-like feature in the upper-left panel of Figure 6, but not so much in the other panels. I suggest maybe overplotting a line or an oval to indicate where the hook-like features are located in the panels.

This comment has become obsolete in the meantime. Due to changes in the training and the bug fix mentioned earlier on, there are now no data points falling in the region of this hook-feature. We have removed the part from the text.

**R1S24)** P18, L13-14: I think it is probably necessary to acknowledge, though, that the presence of rimed particles at the surface indicates that riming is occurring *somewhere* in the column, but that it isn't strong evidence of the correctness of the vertical distribution of FR as predicted by the ANNs.

You are right, we have no ways of telling where riming is happening in the cloud. We propose to add
*"It has to be acknowledged, though, that the vertical distribution of $FR_{ANN}$ cannot be verified by the VISSS observations."*

**R1S25)** P18, L16: Are you saying that you are confident you can apply the ANN #2 to W-band radar observations and get accurate indications of riming?

We want to thank the reviewer for this comment, which led us to overthink our methodology and eventually include the MWACR observations in our training data set. You are correct that we cannot be certain that a method developed on Ka-band radar observations can be transferred to W-band radar observations and reliably produce accurate results. Therefore, we trained one ensemble of ANNs on Ka-band data, and one ensemble of ANNs on W-band data for each of the variable combinations (0, 1 and 2). The TRIPEx-pol data set allows for direct comparison of the two ensembles (see reply to **R1S2**). The predictions look similar for the ANNs trained on Ka-band and W-band observations, however, we think that the methodology is now scientifically more sound.

**R1S26)** P21, L20-25: It seems to me that application to airborne in situ observations (airborne radar plus particle imaging probes) would be a natural extension of this work and could help validate the ANN results for riming aloft.

We want to thank the reviewer for this comment. We agree and propose to add this to the outlook as follows:
*"Further validation, e.g. by comparison of this technique with airborne in situ observations would be a useful extension of this work."*

Technical corrections
######################
P1, L1: SLW has not yet been defined.
P5, L5: "poiting" should be "pointing".
P5, L13: I believe "Snofall" should be "Snowfall".
P5, L25: I believe that "Lee" is not capitalized in this usage.
P7, L18: This is the first use of the term "fully-connected" (the term "fully connected" is used on P8) and the only use of the term "deep neural network". It would be helpful to define them here if they are important to this paragraph.
P10, L13: "17.00" should be "17:00".
P13, L22: "outruled" is more commonly expressed as "ruled out".
P15, Figure 5: Note that the righthand panels are mislabelled as "W-band".
P15, L7-8: Since the first sentence refers to findings in the previous section, it would be clearer to say "the 19 March 2021 Leipzig case".
P15, L9: I think the term "instability" would usually be used, rather than "lability"? Is that the intended meaning?
P16, Figure 6: Please check the arrangement of the panels of the plot. According to the caption, W-band should be on top and Ka-band on the bottom, but this doesn't seem to be the case.

Thanks a lot for the careful reading and for spotting these small errors. We have added all the proposed changes to the updated version of our manuscript.

**Comments by Reviewer #2**

General comment:
The manuscript developed artificial neural networks to estimate rime degrees within ice precipitating clouds using limited data. It is a novel approach to establish ANNs using vertically-pointing radar data. The manuscript well written including detailed enough descriptions of the ANN technique, although I have a few questions about the technique and method. The figures are appropriate. I also have a few questions/concerns that should be clarified before considering publication.

**R2S1)** It was not clear to me why the authors employed the ANN technique rather than a standard technique that explicitly estimates empirical equations e.g. FR(Ze, MDV, SEW, skewness), FR(Ze, SEW, skewness) as Kneifel and Moisseev (2020) derived FR(MDV). I think that deriving such empirical equations is not so hard. Please clarify the advantage of using ANN techniques compared to others.

In the process of deriving a method to detect and quantify riming using cloud radar Dopper moments and SEW, we have tested different approaches, including linear regression, polynomial fitting (as Kneifel & Moisseev (2020) did), random forests, and ANNs. In a first comparison study it became

clear that the ANN method showed the most promising results and we decided to further go down this path.

**R2S2)** A major concern is that this study lacks quantitative evaluations of the established ANNs. The evaluations of the results from the applied ANNs presented in this manuscript are all qualitative, and only a few of short periods near the surface are picked up to compare the results with the in-situ measurements. I understand that the qualitative evaluation of the selected short periods is valuable, but quantitative evaluations, including the entire periods of the application cases, should also be more valuable.

We have added a more thorough evaluation of the test data set, e.g. fitting linear models to the scatterplots of test set targets vs. test set predictions. In addition, the presented RMSEs are a quantitative evaluation metric of the ANNs. Please also see the answers to **R1S18** and **R2S26** on why ANN #0 performs worse than the other two ANN ensembles.

**R2S3)** It was not clear why the authors used SEW instead of Doppler spectrum width. What micriohysical/dynamical processes does SEW represent? How do those processes impact the ANNs? What is the advantage over the Doppler spectrum width? Doppler spectrum width is a standard moment parameter, which is available for even radars that do not collect Doppler spectra, whereas SEW requires Doppler spectrum data, which are expensive for many radars. As the authors pointed near the end of the manuscript, SEW can be strongly influenced by turbulence broadening. Why did the authors use SEW in both ANN1 and ANN2? My specific questions are:
For the significant turbulence cases, if SEW was not used in an ANN, could the result be more reasonable?
Can you quantify the uncertainty attributed to the turbulence broadening in the ANN-based rime degree estimate?
Can you train the ANNs to minimize the turbulence broadening effect?
SEW depends on sensitivity. Do the ANNs work with different systems that have different sensitivity?

We want to thank the reviewer for this remark. There are several points here which need to be addressed.
- Why we use SEW instead of spectrum width: It is correct that spectrum width is a standard moment, and it would be more convenient to use than having to compute the SEW, which in addition requires the storage of the complete Doppler spectra. We did try using spectrum width in an earlier stage of this work, however, ANNs with SEW yielded significantly better results. SEW offers the advantage that it is, opposed to the spectrum width, not weighted by the reflectivity. This means that also a small liquid peak (which can be often observed in Doppler spectra during riming) can lead to an increase in the SEW, whereas spectrum width is not affected as strongly.  Also, we want to point out that this work presents a novel technique, demonstrating how the full potential of Doppler spectra can be used to predict riming.  We acknowledge that we did not make this point clear in our manuscript and propose to add the following sentence to section 2.2 where SEW is explained:
  *"SEW offers the advantage that it is, in contrast to the spectrum width, not weighted by the reflectivity, making it more sensitive to broadening of the spectrum due to small peaks, e.g. caused by liquid water."*
- Impacts of turbulence: Your comment made it clear to us that we need to deal with the impact of turbulence and noise on the ANN predictions somehow. To get a better grasp of the effects of turbulence and the noise level on the ANN performance, we conducted a sensitivity study using Doppler spectra measured by the KAZR radar in Hyytiälä during the BAECC experiment. We

varied the eddy dissipation rate (EDR) and the signal-to-noise ratio (SNR) and simulated the impacts on the Doppler spectra, and the resulting implications for the $FR_{ANN}$ estimate: Spectral broadening was simulated by a convolution of a spectral broadening term $\sigma_T$ with the measured spectrum. $\sigma_T$ can be translated into an EDR using the following relation (e.g. Maahn et al., 2015):

$$\sigma_T^2 = \frac{3\,C}{2} \cdot \left(\frac{\epsilon}{2\,\pi}\right)^{2/3} \left(L_s^{2/3} - L_\lambda^{2/3}\right)$$

Where $\epsilon$ is the EDR, C is the Kolmogorov constant (0.5), and Ls and L$\lambda$ are the largest and smallest length scales observable by the cloud radar, respectively.
The noise floor was varied in a similar manner. Fig. 4 presents the results of this sensitivity study. The left two panels (ANN #0) should not be considered because MDV and Ze do not depend on $\sigma_T$, and SNR also seems to play a minor role. Using the results for the other two ANN ensembles, ANN #1 and #2, we decided to pick $10^{-3}\,m^2 s^{-3}$ as a threshold for the EDR, and SNR=5 as threshold for the signal-to-noise ratio. These two thresholds were used for the training data sampling, the EDR threshold replacing the sampling at cloud base height **(4)**. We propose to change the text describing the training data sampling as follows:

*"During BAECC, a turbulent surface layer was often present, resulting in a broadening of Doppler spectra and impacting their width and other higher-order moments relevant for the training. The following sampling procedure for the training data set was chosen, striving to achieve the best-possible spatio-temporal match between remote sensing and in situ observations, while at the same time avoiding to sample spectra which are strongly impacted by surface induced turbulence: We compute the turbulent eddy dissipation rate (EDR) for 5-minute time periods (Maahn et al., 2015) and determine r, the lowest range at which EDR is $< 10^{-3}\,m^2 s^{-3}$ . This threshold was determined empirically in a sensitivity study, in which spectral broadening was simulated using convolution of a spectral broadening term $\sigma_T$ with measured Doppler spectra."*

In the case studies, where we apply the trained ANNs, we decided to overplot the areas of the cloud with EDR above the selected threshold with grey.

[Figure]

*Figure 4: predicted FR as a function of EDR (top panels) and SNR (bottom panels) for each of the three considered ANN ensembles.*

**R2S4)** The manuscript needs descriptions about how to deal with attenuation effects. Were the radar data corrected for attenuations? The attenuations affect not only reflectivity but also MDV and SEW. Especially, SEW might strongly be affected by the attenuation. Do the attenuations for hydrometeors and gases affect the prediction and the accuracy of the established ANNs?
Especially the Fig.4's case includes many regions of rain, so the attenuation must be significant. Does the attenuation influence the riming prediction above the melting layer?
One of the findings of this study is that the ANNs can be applied at a different cloud radar frequency. Do the attenuation effects influence this?

We want to thank the reviewer for this comment. We have included an attenuation correction for all data sets in our revised analysis **(2)**. We propose to create a new section *2.1.5 Attenuation corrections*, and add the following description:
*"Radar reflectivity of all data used in this study were corrected for attenuation by atmospheric gases, liquid water, melting particles and ice. The Passive and Active Microwave TRAnsfer (PAMTRA) forward operator (Mech et al., 2020) was used in combination with CloudNet products, which are either freely available on the CloudNet data portal (https://cloudnet.fmi.fi) or were processed locally using cloudnetpy (Tukiainen et al., 2020) : Attenuation due to atmospheric gases, including water vapor, was estimated using PAMTRA and the profiles of temperature and humidity included in the CloudNet model files. For liquid water, we used the measured liquid water path and distributed the mass among the pixels classified as "liquid containing" in the CloudNet classification mask, weighted by the measured reflectivity. We used PAMTRA to obtain the attenuation caused by the mass liquid in the respective height bin, assuming a monodisperse particle size distribution for cloud droplets and an exponential distribution for rain (Mech et al., 2020). If pixels were classified as "melting" in*

*CloudNet, the melting layer attenuation was assumed following Matrosov (2008), who derived relations for Ka- and W-band as a function of rainfall rate. Attenuation due to snow and ice was neglected for Ka-band and estimated according to Protat et al. (2019) for the W-band radars. If the cumulative attenuation correction for a pixel exceeded 10 dB, the profile was removed from the analysis."*

**R2S5)** Section 2.2: When I first read this paragraph, it was not clear whether those processes described in this section were applied to all datasets from all sites or only the BAECC datasets. I suppose that they were applied to the BAECC datasets, which were used as training datasets, correct? Please clarify here.

Yes, the training data set only consists of BAECC data.

**R2S6)** p9 line 24: Please describe why the performance of the ANN #0 is much worse than for the others.

Thank you for raising this question. We have addressed this in detail in our response to **R1S18:** The RMSE does not seem to give a strong indication that ANN #0 performs worse than the other two ensembles. We made the decision to exclude ANN #0 after inspecting the RMSE, as well as time-height plots of predicted $FR_{ANN}$ and scatter plots of ANN performance on the test set.  Fig. 2 shows scatter plots (colored by density) for the different ANN ensembles, for both Ka and W-band, on the testing data set.  We are also including a time-height plot of $FR_{ANN}$ for the BAECC riming case in Fig. 3.  It is visible in both the scatter plot and the time-height plot that ANN #0 (trained on MWACR data) seems to only distinguish between two 'modes', low values around 0.4 and higher values around 0.65. The other two set ups are able to yield a range of FR values.

It is obvious that ANN #0 performs "much worse" than the other two ensembles, however, this observation is not reflected by the RMSE. In another attempt to put numbers on this observation, we fit linear models to the scatter plots of the testing data set. The slope should ideally be close to 1. The slopes (red dashed lines in Fig. 2) are listed in Table 1. At both considered frequencies, the results for ANN #0 are lower than for the other two ensembles.

We propose to include Fig. 2 and discuss and justify our decision to let go of ANN #0 in the paper as follows:

*"In the testing phase, the model performance is evaluated using the test set prediction RMSE, in connection with visual inspection of scatter plots of ANN test set predictions and target values. In addition, time-height plots of ANN predictions are examined with regard to physical plausibility of the predicted features. We decided to put an additional focus on the ability of the networks to predict high FR values > 0.5. The reason for this choice is that the extracted Doppler spectra features are expected to contain little information about riming up to a moderate riming stage. For high FR values, the prediction should be more accurate because the riming signal should be clearer in the input features. Table 2 summarizes the test set RMSEs found for the three different ANN input parameter sets and the two radar frequencies. Fig. 2 shows scatter plots of ANN predictions and test set target values, for the three different ANN ensembles, for Ka- and W-band. The slopes of the linear models (red dashed lines in Fig. 2) are listed in Table 2. From Fig. 2 it becomes apparent that ANN #0 seems to have a problem in the W-band set up, because two populations of values are separated in the predictions. The slopes of the linear fits, which should ideally be 1, are lowest for ANN #0 for both considered radar frequencies. This observation is however not reflected in the RMSEs listed in Table 2, which hardly differ between*

*the three ANN ensembles for the W-band. The RMSEs in turn show that ANN #0 performs worse than the other two setups for the Ka-band. This illustrates the limits of using a single quantity like RMSE as a quality metric. ANN #1 and #2 also have issues predicting high FR values, and all ANNs feature a cut-off at low FR values. This might be because riming signatures for low FR are not very pronounced in the cloud radar Doppler spectra, making it impossible for the ANNs to accurately predict FR < 0.3 from the extracted features. Similarly, the issues of the ANNs to predict high FR values could be explained by spectra of pure graupel not necessarily being bimodal, which would result in a less clear signature in SEW and skewness. For the remaining study, we will focus only on ANN #1 and ANN #2"*

**R2S7)** p10 lines 1-2: Please explain more why "riming index" is used instead of FR. What is the relationship between riming index and FR? Please explain more why the output from the training process is different from FR for readers that do not have expertise in ANN.

Referee 1 raised a similar concern in **R1S19**, and we proposed to rename the predicted quantity to $FR_{ANN}$, to make clear that the quantity stays the same. The rime fraction derived from PIP measurements is proposed to be named $FR_{PIP}$.

**R2S8)** Section 3.2: Most of this case seems to have a melting layer, meaning rain near the surface. Which height/time was characterized as "aggregation" or "riming"?

We are referring to Mróz et al. (2020), who defined the "aggregation" and the "riming" periods using a combination of $DWR_{X,Ka}$ , lidar backscatter and MDV. The ranges are approximately 1.5 to 3 km. The times (6:45 to 7:45 for the aggregation case, and a short period around 9:00 UTC) are already mentioned in the text. We propose to shorten the considered case to the time from 2:00 to 11:30 UTC and reorganize the description of the cases as follows to improve comprehensibility:

*"We will first focus in more detail on the 24 November 2018 case from the TRIPEx-pol campaign. This precipitation event has been analyzed with respect to rain and ice microphysics by Mroz et al. (2020), who found strong signatures of aggregation during the period from 06:45 to 07:45 UTC, and riming during a shorter time interval around 09:00 UTC.*
*Fig. 5 shows the radar moments measured during the period from 02:00 UTC to 11:30 UTC on 24 November 2018. The period characterized as "aggregation'' by Mróz et al. (2020) clearly shows up as a patch of increased signal in the $DWR_{X,Ka}$ between 1 and 4 km range, while during the `"riming" period, an obvious increase in absolute MDV values can be observed in a similar range interval, along with an increase in SEW."*

**R2S9)** Figure 6: Many data points with higher riming index > 0.7 are deviated from the pink line. Are they accurate predictions? I think that adding the same plots but with color scales with FR will be useful.

The plot has been updated **(5)(6)**, and now most of the points fall very closely to the pink line.

**R2S10)** p21 line 19: In cases where MDV is not available due to orographic effects, SEW could also be impacted by a turbulence.

We are hoping to counteract this matter by masking out pixels with high EDR. We are doing this in all time-height plots of predicted FRANN, using the EDR threshold $10^{-3}$ $m^2s^{-3}$ **(3)**

Technical comments:

p4, line 10: SEW is first used here.
Table 1: Please give the total time of the datasets used. This would also be useful.
p9 line2: I am not sure if five folds are enough for the validation process. Could you evaluate/explain that the number is appropriate?

The number of folds has in the meantime been changed to three. This choice is to some degree subjective, and was made after evaluation of the validation data set variance.

P10 line 18: What is the unit of 0.7 and 0.9?
kilometers. We added the unit here. Thanks for spotting this.

Figure 2: Please add temperature contours similar to Fig. 5.
Done

Figure 3: Please add time series of FR estimated from the in-situ measurements and compare it with the riming index from the ANNs.
We added the suggested panel. However, it should be noted that there is a gap in the observations just around the time when the most prominent riming happens aloft. We mentioned this in the previous version of our manuscript: *"Unfortunately, due to low precipitation intensity at the ground, no PIP-based FR retrieval was available for this time period."*

Were DWRs corrected for attenuations?
Yes.

p13 line 15: When is the riming period?
We have added "around 09:00 UTC"

4d: Please use appropriate color scale.
We changed the limits of the color scale.

Figure 5: Is it possible to have riming with T<-40C (0.4 riming index)?
The figure has changed and now no increased riming at these lower temperatures is visible.

Legends on Figs. 5a-5d present all "W band."
The labels have been adjusted.

Figure 6: I think that the four panels in this figure used the same dataset, and the distribution of the data points should be same. Why is the distribution in the plots in the left column and the right column different?
The left column is W-band and the right column is Ka-band.

Figure 8: Could you add time series of FR estimated from the in-situ measurements or similar parameter from the image analysis?
Unfortunately, no. The VISSS instrument is a very new development and products such as estimated FR still have to be established.

---

## Author Response (AR2)

We want to thank the editor and both anonymous referees again for reviewing the changes made to our manuscript. We are addressing their comments in a point-by-point way below. In addition, we want to propose one more change to the abstract of our paper.

**Change to the Abstract: We noticed that the abbreviation FR$_{ANN}$ is not properly introduced. We propose to change the relevant part of the text as follows:**

*"Training data are extracted from a data set acquired during winter 2014 in Finland, containing both Ka- and W-band cloud radar and in-situ observations of snowfall by a Precipitation Imaging Package, from which the rime mass fraction (FR$_{PIP}$) is retrieved. ANNs are trained separately either on the Ka-band radar or the W-band radar data set to predict the rime fraction FR$_{ANN}$."*

**Referee #1: I have very minor comments as follow. The page numbers and line numbers refer to the authors' tracked changes document, amt-2021-137-ATC1.pdf:**

**P10, L8-9: Should this say "...while the remaining two folds..."?**

You are right, thanks for spotting this error. We changed the text accordingly.

**P11, Figure 2 caption: Capitalize "scatter".**

Done.

**P12, L11-12: Perhaps place this statement defining FR_ANN a bit earlier in the text, around P10 L33-34, with the initial discussion of Figure 2.**

Good point, we moved the sentence *"We will refer to the predicted quantity in the following as FR$_{ANN}$, to distinguish it from the FR$_{PIP}$ retrieved from in-situ observations"* a little up in the text.

**P16, L27-29: For clarity in this statement about the correlations of FR_ANN values, could you add information about what variable(s) the FR_ANN values are being correlated against?**

Thanks for pointing us to this poorly understandable sentence. We changed it as follows:

*"The correlation of the predicted FR$_{ANN}$ values for all considered TRIPEx-pol cases is high for both ANN sets, the R² of the Ka- vs. W-band-based predictions being 0.73 for ANN #1 and 0.81 for ANN #2, respectively."*

**Referee #2: I am satisfied with the authors revision and responses. My additional comment is on attenuation correction: According to the authors, attenuation correction for liquid is mandatory particularly for shorter wavelength radars. Please also give short comments about attenuation effects by supercooled liquid droplets (including the limitation impacting the technique), because this study focused on riming.**

We added the following sentence to the description of the attenuation correction to make this more clear:
*"This means that attenuation caused by SLW droplets is only corrected for in SLW layers detected by the CloudNet classification mask, i.e. limited by lidar signal attenuation."*